# Intrasexual cuticular hydrocarbon dimorphism in a wasp sheds light on hydrocarbon biosynthesis genes in Hymenoptera

Victoria C. Moris [1,2 ✉], Lars Podsiadlowski [3,4], Sebastian Martin [3,4], Jan Philip Oeyen [3,5], Alexander Donath [3], Malte Petersen [3,6], Jeanne Wilbrandt [3,7], Bernhard Misof [3], Daniel Liedtke [8], Markus Thamm [9], Ricarda Scheiner [9], Thomas Schmitt [10] & Oliver Niehuis [1 ✉]

Cuticular hydrocarbons (CHCs) cover the cuticle of insects and serve as desiccation barrier and as semiochemicals. While the main enzymatic steps of CHC biosynthesis are well understood, few of the underlying genes have been identified. Here we show how exploitation of intrasexual CHC dimorphism in a mason wasp, *Odynerus spinipes*, in combination with whole-genome sequencing and comparative transcriptomics facilitated identification of such genes. RNAi-mediated knockdown of twelve candidate gene orthologs in the honey bee, *Apis mellifera*, confirmed nine genes impacting CHC profile composition. Most of them have predicted functions consistent with current knowledge of CHC metabolism. However, we found first-time evidence for a fatty acid amide hydrolase also influencing CHC profile composition. In situ hybridization experiments furthermore suggest trophocytes participating in CHC biosynthesis. Our results set the base for experimental CHC profile manipulation in Hymenoptera and imply that the evolutionary origin of CHC biosynthesis predates the arthropods' colonization of land.

[1] Department of Evolutionary Biology and Ecology, Institute of Biology I (Zoology), University of Freiburg, 79104 Freiburg, Germany. [2] Laboratory of Molecular Biology & Evolution (MBE), Department of Biology, Université Libre de Bruxelles, 1000 Brussels, Belgium. [3] Centre for Molecular Biodiversity Research, Leibniz Institute for the Analysis of Biodiversity Change / ZFMK, Museum Koenig, Adenauerallee 160, 53113 Bonn, Germany. [4] Institute of Evolutionary Biology and Ecology, University of Bonn, An der Immenburg 1, 53121 Bonn, Germany. [5] Centre for Ecological and Evolutionary Synthesis, Department of Biosciences, University of Oslo, NO-0316 Oslo, Norway. [6] High Performance Computing & Analytics Lab, University of Bonn, Friedrich-Hirzebruch-Allee 8, 53115 Bonn, Germany. [7] Leibniz Institute on Aging — Fritz Lipmann Institute, Beutenbergstraße 11, 07745 Jena, Germany. [8] Institute of Human Genetics, University of Würzburg, Am Hubland, 97074 Würzburg, Germany. [9] Department of Behavioral Physiology and Sociobiology, University of Würzburg, Am Hubland, 97074 Würzburg, Germany. [10] Department of Animal Ecology and Tropical Biology Biocenter, University of Würzburg, Am Hubland, 97074 Würzburg, Germany. ✉email: victoria.carla.moris@gmail.com; oliver.niehuis@biologie.uni-freiburg.de

Cuticular hydrocarbons (CHCs) are long-chained molecules that cover the cuticle of most, if not all, insects[1]. They function as important desiccation barrier and are thought to have fostered the transition from marine to terrestrial habitats in the lineage leading to Hexapoda[2]. CHCs additionally serve as semiochemicals for inter- and intraspecific communication[2]. In Hymenoptera, they play a particularly important role in establishing eusociality by enabling kin, nest-mate, and caste recognition[2]. CHCs have also been exploited for chemical deception and chemical mimicry[2]. Identifying genes involved in CHC biosynthesis of Hymenoptera is of major interest, as it would open the door for CHC profile manipulation (e.g., via CRISPR-Cas9 or RNA interference) in behavioral, ecological, and socio-biological studies on Hymenoptera.

Most insects synthesize CHCs de novo from fatty acid precursors[3] in oenocytes[4]. These secretory cells of ectodermal origin are found in the fat body of all major insect lineages[4]. The main enzymatic steps of CHC biosynthesis are well understood and involve fatty acid synthases (FAS), fatty acid elongases (ELO), fatty acid desaturases (Desat), fatty acyl-CoA reductases (FAR), and cytochrome p450 decarbonylases (CYP4G subfamily)[5–10] (Fig. 1). However, the details of CHC biosynthesis and which genes encode the participating enzymes are not yet fully elucidated, particularly in the species-rich insect order Hymenoptera, in which only a single gene (CYP4G) has so far been shown to be involved in CHC biosynthesis[10,11]. Furthermore, insights obtained from studying species of one insect lineage (e.g., Diptera, in which three genes coding for fatty acid desaturases have been shown to be involved in alkene synthesis based on research on *Drosophila*[10]) are not necessarily representative for species of a distantly-related lineage (e.g., Hymenoptera). Most genes associated with CHC biosynthesis belong to multi-gene families, and some of these exhibit remarkably high turnover rates (e.g., fatty acid desaturases[12]). Therefore, it is highly desirable to identify CHC biosynthesis-related genes in multiple insect lineages. A promising approach to this end is transcriptomic analysis comparing gene expression between samples of the same species that differ in their CHC composition from each other.

In most species, individuals of the same sex share the same set of CHCs, which is referred to as CHC profile type[13]. Intriguingly, previous research found that females of the mason wasp *Odynerus spinipes* are able to express two CHC profile types[14]. The two chemotypes (c1 and c2) qualitatively differ in 77 CHCs from each other, with most of the differing CHCs being alkenes[14]. Specifically, chemotype 1 is characterized by alkenes with double bonds at uneven positions, whereas chemotype 2 is characterized by alkenes with double bonds at even positions[14,15]. Furthermore, wasps expressing chemotype 2 feature a higher relative abundance of alkenes with chain lengths greater than 27 carbon atoms than wasps expressing chemotype 1[14,15] (Supplementary Fig. 1). The CHC dimorphism of the spiny mason wasps provides an exceptional opportunity to identify genes involved in CHC biosynthesis, as *O. spinipes* females with different chemotypes do not appear to differ in any other trait from each other; hence, they should systematically differ only in the expression of CHC biosynthesis-related genes from each other[14,15]. *O. spinipes* females additionally show age-related quantitative changes in their CHC profile composition, which can also be used to identify candidate genes.

In this study, we report results from exploiting the CHC dimorphism in the spiny mason wasp for identifying CHC biosynthesis-related candidate genes in Hymenoptera and testing the functional involvement of the candidate genes in CHC biosynthesis in the honey bee — a more trackable species of Hymenoptera for reverse genetic analyses than the mason wasp. We first tested whether *O. spinipes* females retain a given chemotype throughout their adult life. We concurrently sequenced, assembled, and annotated the genome of *O. spinipes* to aid accurate analysis of gene expression levels in this species. We next identified candidate genes differentially expressed between *O. spinipes* females with different CHC profile composition. We then identified candidate gene orthologs in the *Apis mellifera* genome to facilitate reverse genetic analyses in species that allows us to study a larger number of replicates per knockdown experiment than *O. spinipes*. Specifically, we investigated which cell types of the worker honey bee fat body express the candidate genes and tested whether knockdown of selected candidate genes in worker honey bees via RNA interference (RNAi) impacts their CHC profile composition. These experiments allowed us to identify an array of genes impacting the honey bee's CHC profile composition and provided hints what functions the encoded enzymes might have in the CHC biosynthesis of this species and of Hymenoptera in general.

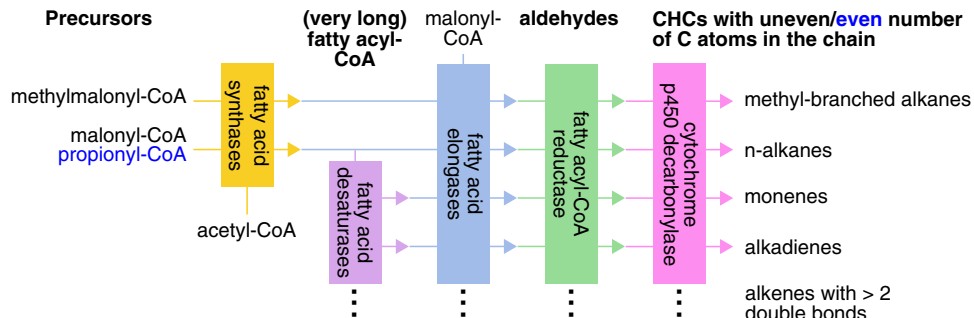

**Fig. 1 Schematic overview of the main steps of cuticular hydrocarbon biosynthesis.** The pathway starts with a fatty acid synthase (FAS) that binds acetyl-coenzyme A (CoA) with malonyl-CoA units. The resulting acyl-CoA (with an even number of carbon atoms) is repeatedly elongated by a FAS which binds acyl-CoA with malonyl-CoA. During each elongation step, the chain length of the fatty acyl-CoA increases by two carbon atoms. Fatty acyl-CoA chains with an uneven number of carbon atoms are thought to arise from substituting malonyl-CoA with propionyl-CoA during the elongation step. The consecutive elongation of the fatty acyl-CoA chain by FAS results in long chain fatty acyl-CoAs. Additional elongation of these fatty acyl-CoA chains to very-long fatty acyl-CoA requires the action of fatty acid elongases that add additional malonyl-CoA molecules to the long chain fatty acyl-CoA. These are reduced into aldehydes by fatty acyl-CoA reductases. Before the latter step, double bonds can be inserted in the (very-)long chain fatty acyl-CoAs by fatty acid desaturases, resulting in the biosynthesis of unsaturated hydrocarbons with a different number of double bonds (alkenes, alkadienes). The biosynthesis of methyl-branched alkanes is thought to depend on the action of a specialized fatty acid synthase that incorporates methylmalonyl-CoA instead of malony-CoA into the chain at specific positions during the chain elongation. The last step of CHC biosynthesis is the oxidative decarbonylation of aldehydes leading to hydrocarbons (with the loss of one carbon atom) by a cytochrome p450. Based on[2,5-9].

**Table 1 Cuticular hydrocarbon biosynthesis candidate genes in the mason wasp *Odynerus spinipes*.**

| Wasp gene ID | DGE between | Honey bee gene ID | %ID | Predicted function | Cell type |
|---|---|---|---|---|---|
| g3158 | age classes | GB52590 | 54.7 | fatty acid synthase | oenocytes (weak) |
| g6537 | age classes | GB46038 | 85.9 | fatty acid elongase | oenocytes + trophocytes |
| g7609 | age classes | GB54399 | 58.7 | fatty acid elongase | oenocytes (weak) |
| g7610 | age classes | GB54397 | 55.2 | fatty acid elongase | oenocytes |
| g7616 | age classes / chemotypes | GB51247 | 82.9 | fatty acid elongase | oenocytes |
| g7620 | age classes | GB54401 | 84.0 | fatty acid elongase | trophocytes |
| g7621 | age classes | GB54302 | 67.7 | fatty acid elongase | NA |
| g14708 | age classes | NA[a] | NA[a] | fatty acid desaturase | oenocytes |
| g14712 | chemotypes | GB42218 | 54.2 | fatty acid desaturase | oenocytes |
| | | GB51236 | 64.1 | | oenocytes |
| | | GB51238 | 55.2 | | trophocytes |
| g2413 | age classes | GB50627 | 64.2 | fatty acyl-CoA reductase | oenocytes |
| g1571 | age classes | GB52087 | 48.1 | fatty acyl-CoA reductase | oenocytes + trophocytes |
| g7842 | age classes | GB49380 | 72.3 | fatty acyl-CoA reductase | oenocytes |
| g283 | chemotypes | GB44756 | 36.9 | rhythmically expressed gene 5 protein | NA |
| g2290 | chemotypes | GB53695[b] | 34.3 | fatty acid amide hydrolase | oenocytes + trophocytes (weak) |
| g3059 | chemotypes | GB52820 | 62.4 | parathyroid hormone-related peptide receptor-like protein | oenocytes (weak) + trophocytes (weak) + hexagonal cells (strong) |

The table informs about 1) the mason wasp gene ID, 2) between which groups differential expression differences (DGE) were detected, 3) orthologous/homologous genes in the honey bee (*Apis mellifera*) genome, 4) the percentage of identical amino acids between the encoded sequences of the mason wasp and those of the honey bee, 5) the genes' predicted functions, and 6) the expression of the candidate genes in honey bee fat body cells.
[a]Since we found no 1:1 ortholog in the honey bee genome, we assessed the phylogenetically most closely related homolog, GB40659 (39.6%), for being involved in CHC biosynthesis.
[b]One of three co-orthologs in the honey bee, the other two being GB48850 (36.4%) and GB47832 (34.3%) (not studied by us).

## Results

**Odynerus spinipes chemotype persistency**. None of the *O. spinipes* females whose CHC profile we sampled multiple times during their adult life changed their chemotype (c1 and c2): neither those kept under laboratory conditions ($N_{c1}$ = 10; $N_{c2}$ = 17; sampled on average three times during the first 16 days after their eclosion; Supplementary Table 1), nor those studied in field populations ($N_{c1}$ = 8; $N_{c2}$ = 10; sampled on average two times and with a time difference of on average five days; Supplementary Table 2). These results are consistent with the hypothesis that adult *O. spinipes* females cannot change their chemotype and justify associating CHC chemotype differences between females with gene expression differences between these females.

**Odynerus spinipes draft genome**. To foster comparative transcriptomic analyses on *O. spinipes*, we sequenced the genome of this species using short read sequencing technology. The assembly of the *O. spinipes* genome resulted in 21,922 scaffolds spanning a total of 171 Mbp, 89.5% of the total genome, which we estimated via flow cytometry to be ca. 191 Mbp. The N50 value and L50 value of the draft genome assembly are 3.6 Mbp and twelve scaffolds, respectively. Exploiting whole body transcript libraries of one adult male and of two adult females (representing each chemotype) to foster annotation of nuclear-encoded protein-coding genes (Supplementary Table 3), we inferred a total of 16,677 gene models. The nucleotide sequence data of the genome shotgun project are deposited under the NCBI Bioproject PRJNA735081. The draft genome assembly and the corresponding gene annotations are available under the NCBI accession JAIFRP000000000 and from Zenodo (10.5281/zenodo.7334422).

**Candidate genes in *Odynerus spinipes***. We first identified statistically significant differentially expressed genes in transcriptomes of females with different chemotype in two batches of adult females that differed in sampling time: 12–38 h (batch 1) and 48–62 h (batch 2) after the wasps' eclosion (Supplementary Tables 4–6).

Considering only genes that were consistently differentially expressed between chemotypes in the two batches, we retained five candidate genes (Table 1; Supplementary Tables 7 and 8). We next searched for genes that were significantly differentially expressed between the two above batches of adult female wasps that differed in their age from each other, ignoring the wasps' chemotype. However, we only considered genes belonging to four gene families known to include genes involved in CHC biosynthesis (i.e., FAS, ELO, Desat, FAR) based on prior research[10]. This procedure resulted in the identification of eleven candidate genes (Table 1; Supplementary Table 9).

Combining the results from both approaches and accounting for redundancy (*g7616* was identified with both search strategies), we identified a total of 15 candidate genes affecting CHC biosynthesis in *O. spinipes* (Table 1). Based on amino acid sequence homology to proteins whose functions have been characterized in other species before, the candidate genes encode one fatty acid synthase, six fatty acid elongases, two fatty acid desaturases, three fatty acyl-CoA reductases, one fatty acid amide hydrolase, one parathyroid hormone-related peptide receptor-like protein, and one rhythmically expressed gene 5 protein.

**Honey bee candidate gene orthologs**. We applied a phylogenetic approach to identify orthologs of twelve candidate genes belonging to multi-gene families (i.e., FAS, ELO, Desat, FAR) in the honey bee genome. The majority (ten) of the *O. spinipes* candidate genes share a 1:1 ortholog relationship to honey bee genes (Figs. 2 and 3; Supplementary Figs. 2 and 3; Table 1). However, the fatty acid desaturase candidate gene (*g14712*) of *O. spinipes* was found to have three co-orthologs in *A. mellifera*. We decided to consider all of them for further analyses (Table 1; Fig. 3). We found one fatty acid desaturase candidate gene (*g14708*) to not have an ortholog in *A. mellifera*. We decided to include the phylogenetically most closely related *A. mellifera* gene for further experimental assessment (Table 1; Fig. 3).

We applied the best reciprocal hit criterion to identify orthologs of the remaining three candidate genes (*g283, g2290, g3059*) in the

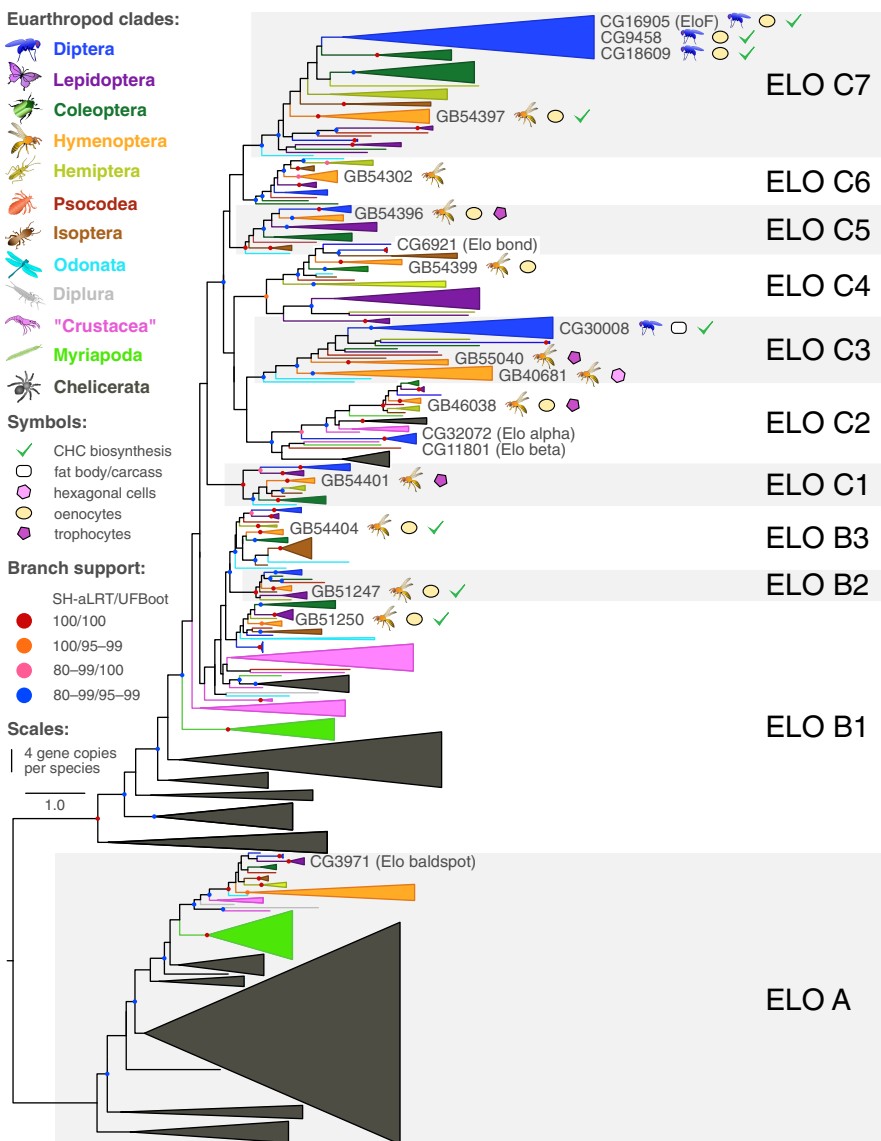

**Fig. 2 Gene tree of fatty acid elongases from 37 species of Euarthropoda showing copy numbers and involvement of genes in CHC biosynthesis.** Branch support was calculated from ultrafast bootstrap (UFBoot2) replicates (in %) and Shimodaira-Hasegawa approximate likelihood ratio tests (SH-aLRT in %). Supported branches are represented by a color circle, whereas those with SH-aLRT<80% and UFBoot<95% do not have a circle. The width of each clade represents the number of gene copies per species (single copies represented by a line).

honey bee genome (Table 1). In all instances except one did we find 1:1 orthologs. The exception was a gene predicted to encode a fatty acid amide hydrolase (FAAH) (*g2290*) that we found to be represented by three co-orthologs in the honey bee genome. Because this gene family had previously not been implicated with CHC biosynthesis, we considered only one ortholog (*GB53695*), that with the best reciprocal hit to *g2290* (i.e. with the smallest e-value), in subsequent experiments.

We extended our list of candidate genes by seven genes (Table 2). Two of them are *A. mellifera* orthologs of genes previously shown to be involved in the CHC biosynthesis in the fruit fly *Drosophila melanogaster* (i.e., the honey bee ortholog of the two fatty acid synthases *FASN1*[16]/*FASN2*[9], Supplementary Fig. 2, and the honey bee ortholog of the three desaturases *Desat1*[17], *Desat2*[17], and *DesatF*[18] (Fig. 3). The remaining five genes encode fatty acid elongases that belong to clades phylogenetically closely related to genes shown in previous studies (*CG30008*[19], *EloF*[20]) and in this study to be involved in CHC biosynthesis (Fig. 2). The total number

of candidate genes selected for further investigation in *A. mellifera* was thus 24 (Tables 1 and 2).

**Candidate gene expression sites.** In situ hybridization (ISH) experiments on fat body tissues from honey bee metasomas revealed 18 of the 24 candidate genes to be expressed in oenocytes: a fatty acid synthase (*GB52590*), eight fatty acid elongases (*GB46038*, *GB54396*, *GB54397*, *GB51250*, *GB51247*, *GB54399*, *GB54404*, *GB55040*), four fatty acid desaturases (*GB40659*, *GB42218*, *GB48195*, *GB51236*), three fatty acyl-CoA reductases (*GB49380*, *GB50627*, *GB52087*), one fatty acid amide hydrolase (*GB53695*), and one parathyroid hormone-related peptide receptor-like protein (*GB52820*) (Fig. 4; Tables 1 and 2; negative controls performed with sense probes are shown in Supplementary Fig. 4). Six of them (three fatty acid elongases, one fatty acid desaturase, one fatty acyl-CoA reductase, one fatty acid amide hydrolase) were found to be co-expressed in trophocytes, a

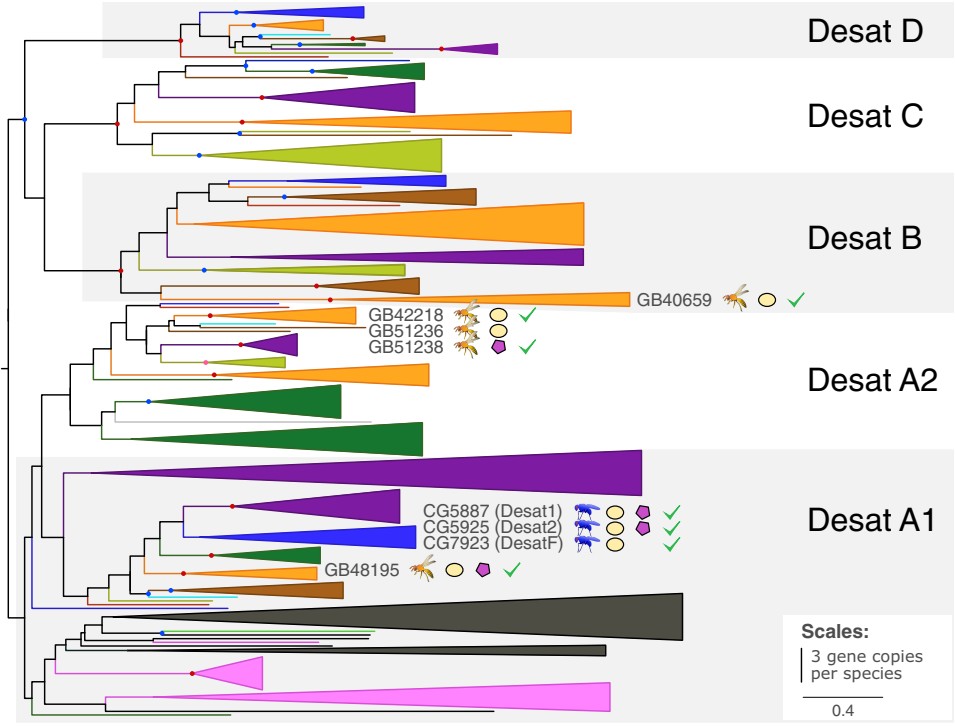

**Fig. 3 Gene tree of fatty acid desaturases from 37 species Euarthropoda showing copy numbers and involvement of genes in CHC biosynthesis.** Color coding and symbols are as in the legend of Fig. 2.

**Table 2 Additional cuticular hydrocarbon biosynthesis candidate genes studied in the honey bee (*Apis mellifera*).**

| Fly gene ID | Honey bee gene ID | Predicted function | Cell type |
|---|---|---|---|
| FASN1/2 | GB53412 | fatty acid synthase | hexagonal cells |
| CG30008 | GB55040 | fatty acid elongase | oenocytes + trophocytes |
| CG30008 | GB40681 | fatty acid elongase | hexagonal cells |
| CG5326 | GB54396 | fatty acid elongase | oenocytes + trophocytes |
| CG31522 | GB51250 | fatty acid elongase | oenocytes |
| CG2781 | GB54404 | fatty acid elongase | oenocytes |
| Desat1, 2, and F | GB48195 | fatty acid desaturase | oenocytes + trophocytes |

Given are 1) the fly and 2) the honey bee gene IDs, 3) the genes' predicted functions, and 4) the expression of the candidate genes in honey bee fat body cells.

fat body cell type different from oenocytes (Fig. 4; Tables 1 and 2). We found two candidate genes, one fatty acid desaturase (*GB51238*) and one fatty acid elongase (*GB54401*), to be expressed in trophocytes only (Fig. 4; Table 1). Three candidate genes, predicted to encode a fatty acid synthase (*GB53412*), a fatty acid elongase (*GB40681*), and a parathyroid hormone-related peptide receptor-like protein (*GB52820*), respectively, were found to be expressed in cells located between the cuticle and the fat body layer (*GB52820* was found also slightly expressed in oenocytes and trophocytes). We refer to these cells as hexagonal cells (Fig. 4; Tables 1 and 2). In situ hybridization experiments on one candidate gene, predicted to encode a rhythmically expressed protein (*GB44756*), were inconclusive (marked by an asterisk in Fig. 4), because the staining intensity of the treatment group did not differ from that of the negative controls (Supplementary Fig. 4). We were unable to study cell-specific expression of one candidate gene (*GB54302*) encoding a fatty acid elongase, as its probe synthesis failed.

**Candidate gene knockdown experiments.** We assessed twelve candidate genes (selection criteria given in Supplementary Methods)

for influencing CHC profile composition by conducting RNAi-mediated knockdown experiments on worker honey bees (Figs. 5 and 6). Knockdown of nine candidate genes resulted in statistically significant changes of the CHC profile composition of the treated bees relative to control experiments in which we injected double-stranded RNA (dsRNA) of green florescent protein (GFP) (powered partial least squares discriminant analysis [PPLS-DA]: $p \leq 0.05$ after Benjamini-Hochberg correction for multiple testing). The nine genes were: four fatty acid elongases (*GB51250*, *GB51274*, *GB54397*, *GB54404*) (Fig. 5), four fatty acid desaturases (*GB40659*, *GB42218*, *GB48195*, *GB51238*) (Fig. 6), and one fatty acid amide hydrolase (*GB53695*) (Fig. 5). CHC profile compositional changes associated with the knockdown of fatty acid desaturase *GB51236* were statistically insignificant after Benjamini-Hochberg correction for multiple testing (PPLS-DA, $p = 0.087$; Supplementary Fig. 5). RNAi-mediated knockdown of the remaining two genes, a fatty-acid synthase (*GB52590*) and a fatty-acyl-CoA reductase (*GB52087*), caused a premature death of the treated bees within the first two days after dsRNA injection.

While we found RNAi of the nine genes outlined above to generally result in a strong phenotypic effect 2–5 days after

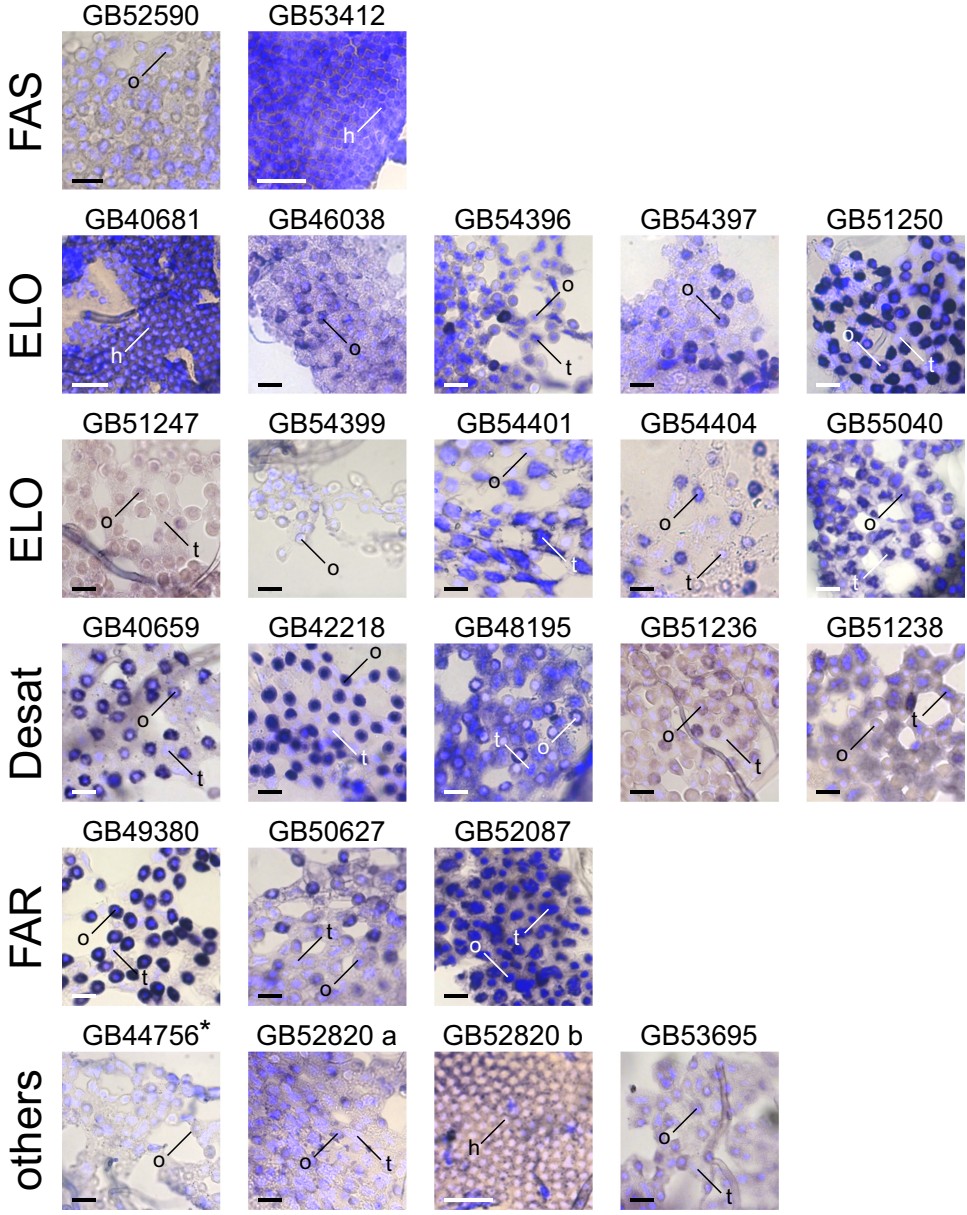

**Fig. 4 Cell-specific expression of candidate genes in the fat body of worker honey bee (*Apis mellifera*).** The photomicrographs visualize the expression (dark blue/purple) of fatty acid synthases (FAS), fatty acid elongases (ELO), fatty acid desaturases (Desat), fatty acid reductases (FAR), and additional candidate gene transcripts (others) in fat body cells. Three fat body cell types are indicated: hexagonal cells (h), oenocytes (o), and trophocytes (t). Two separate photomicrographs visualize the expression of *GB52820*. DAPI (in light blue) was used to counterstain nuclei. Staining in the photomicrograph for *GB44756* (marked with an asterisk) did not significantly differ from that of the corresponding control (Supplementary Fig. 4). Scale bar: 50 μm.

dsRNA injection, we did not detect statistically significant differences in expression levels of most target genes at the same point in time ($p = 0.126–0.607$; Welch's t-tests; Figs. 5 and 6). The exceptions are three fatty acid desaturases (*GB42218*, *GB48195*, *GB51238*) that showed significantly lower expression levels in treated bees relative to control bees after Benjamini-Hochberg correction for multiple testing (*GB42218*: Wilcoxon signed-rank test, $W = 122$, $p = 0.002$; *GB48195*: Welch's t-test, $t_{(10.85)} = 4.01$, $p = 0.010$; *GB51238*: Welch's t-test, $t_{(9.67)} = 3.21$, $p = 0.033$; Fig. 6). Note that we were unable to measure the expression levels of GB54397 (encoding a fatty acid elongase) and of GB40659 (encoding a fatty acid desaturase), because the qRT-PCR oligonucleotide primers did not efficiently amplify a unique stretch of the genes' cDNA.

PPLS-DA of the CHC profiles provided information on the compounds that dominantly discriminate the RNAi treatment group and the control group. Knockdown of fatty acid elongases caused major CHC profile perturbations in several compound classes (Supplementary Fig. 6), but it most notably impacted CHC chain lengths (Fig. 5). Specifically, we found the chain lengths of CHCs positively correlating more than 50% along the first principal component of the PPLS-DA (Fig. 5) with a fatty acid elongase dsRNA treatment (*GB51247*, *GB51250*, *GB54397*, *GB54404*) to be significantly lower than CHC chain lengths positively correlating with a control dsRNA treatment after correcting for multiple testing (Holm-Bonferroni) (*GB51247*: Welch's t-test, $t_{(12.04)} = 6.67$, $p < 0.001$; *GB51250*: Welch's t-test, $t_{(15.98)} = 6.18$, $p < 0.001$; *GB54397*: Welch's t-test, $t_{(27.76)} = 3.54$, $p = 0.003$; *GB54404*: Welch's t-test, $t_{(24.93)} = 2.58$, $p = 0.016$) (Fig. 7). We found the opposite pattern when interfering with the expression of GB53695 (encoding a fatty acid amide hydrolase): the CHC profiles of the treatment bees contained CHCs with

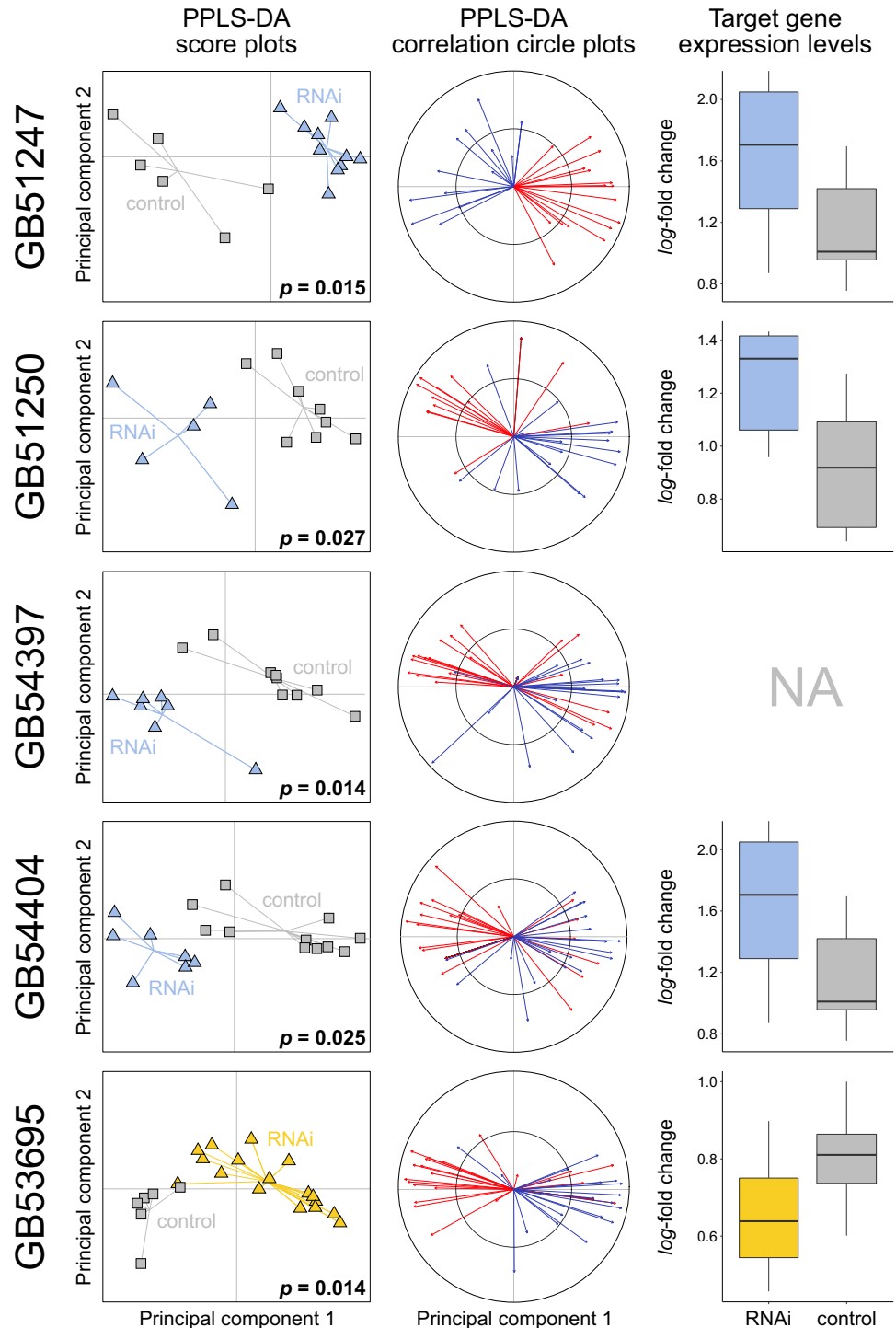

**Fig. 5 Effects of RNAi-mediated knockdown of four fatty acid elongases and of one fatty acid amide hydrolase (FAAH) in worker honey bees (*Apis mellifera*) 2–5 days after dsRNA treatment.** The first two columns show score plots and correlation circle plots from a powered partial least squares discriminant analysis (PPLS-DA) of the cuticular hydrocarbon (CHC) profile data of target gene-treated (in blue for fatty acid elongases RNAi: *GB51247, GB51250, GB54397, GB54404*, in yellow for FAAH RNAi: *GB53695*) and of control bees (in grey, injected with dsRNA targeting GFP). *P*-values indicate the statistical probability of group differences representing random variation after Benjamini-Hochberg correction for multiple (N = 10) testing. The correlation circle plots indicate how many CHCs of a given chain length class (i.e., ≤ 27 [red] or > 27 [blue] carbon atoms) correlate with the first two principal components. Box plots in the third column show gene expression levels ($log_2$-fold change) of the target genes in target gene-treated (in blue for elongase RNAi, in yellow for FAAH RNAi) and in control bees (in grey). None of the expression levels differed statistically significantly between target gene-treated and control bees after applying Benjamini-Hochberg correction for multiple testing (N = 9; Welch's t-test and Wilcoxon signed-rank test). Additional information is shown in Supplementary Fig. 6.

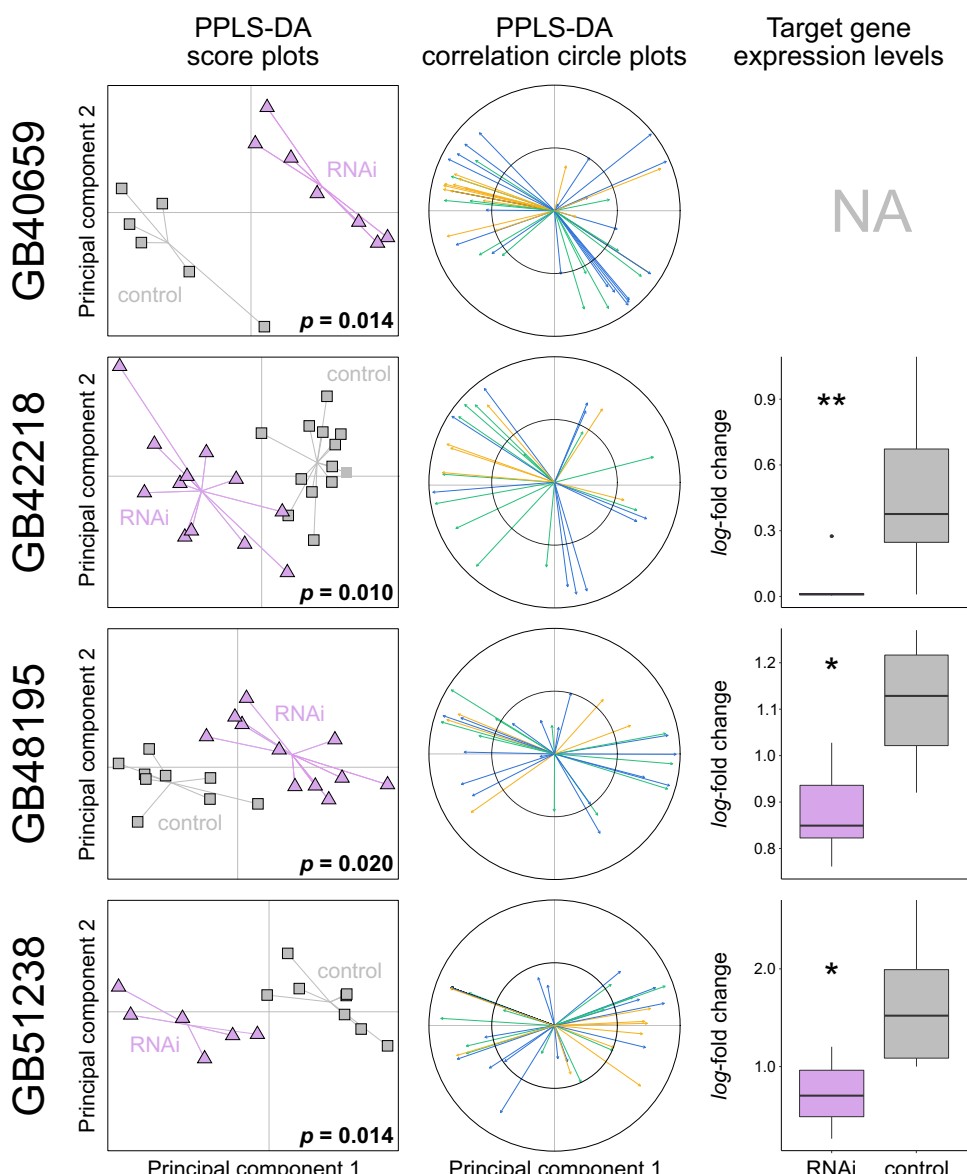

**Fig. 6 Effects of RNAi-mediated knockdown of four fatty acid desaturases in worker honey bees (*Apis mellifera*) 2–5 days after dsRNA treatment.** The first two columns show score plots and correlation circle plots from a powered partial least squares discriminant analysis (PPLS-DA) of cuticular hydrocarbon (CHC) profile data of target gene-treated (in purple: *GB40659, GB42218, GB48195, GB51238*) and of control bees (in grey, injected with dsRNA targeting GFP). *P*-values indicate the statistical probability of group differences representing random variation after Benjamini-Hochberg correction for multiple ($N = 10$) testing. The correlation circle plots indicate how many CHCs of a given compound class (i.e., alkanes [green], alkenes/alkadienes [blue], methyl-branched alkanes [yellow]) correlate with the first two principal components. Box plots in the third column show gene expression levels ($log_2$-fold change) of the target genes in target gene-treated (in purple) and in control bees (in grey). Asterisks indicate the statistical significance of expression differences between target gene-treated and control bees after applying Benjamini-Hochberg correction for multiple testing ($N = 9$; Welch's t-test/ Wilcoxon signed-rank test; $p \leq 0.05$ [*], $p \leq 0.01$ [**]). Additional information is shown in Supplementary Fig. 7.

significantly longer chain lengths than the CHC profiles of the control bees (Welch's t-test, $t_{(27.71)} = -5.92$, $p < 0.001$; Figs. 5 and 7; Supplementary Fig. 6).

Knockdown of fatty acid desaturases also caused major CHC profile perturbations and impacted the relative abundance of several compound classes (i.e., alkanes, alkenes, and methyl-branched alkanes) (Fig. 6; Supplementary Fig. 7). Depending on what gene we targeted with dsRNA, we found statistically significant differences in the relative abundance of alkenes with specific double bond positions between treated bees and control bees. Specifically, we found the relative abundance of alkenes with a double bond at even positions (8,10) to be significantly smaller in bees treated with dsRNA of *GB48195* (Wilcoxon signed-rank test,

$W = 87$, $p = 0.022$) and of *GB51238* (Welch's t-test, $t_{(11.94)} = 5.31$, $p = 0.002$) than in control bees after correcting for multiple testing (Holm-Bonferroni) (Fig. 8). We did not study the impact of target gene knockdown on alkadienes (hydrocarbons with two double bonds), since the relative abundance of alkadienes was close to the detection limit of the used GC-MS instruments.

## Discussion

Studying intrasexual CHC profile differences in the mason wasp *O. spinipes* provided promising candidate genes that modulate CHC profile composition in Hymenoptera. Knockdown experiments on twelve selected candidate genes in *A. mellifera* resulted

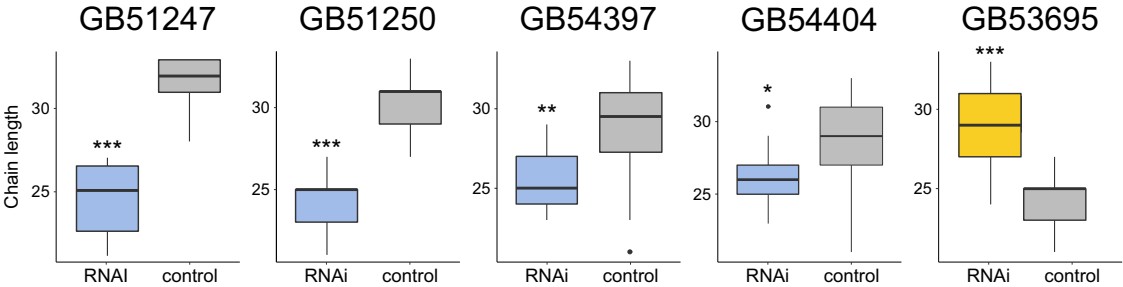

**Fig. 7 Effects on chain lengths of cuticular hydrocarbons (CHCs) of RNAi-mediated knockdown of four fatty acid elongases and of one fatty acid amide hydrolase (FAAH) in worker honey bees (*Apis mellifera*) 2–5 days after dsRNA treatment.** Box plots of the chain lengths of CHCs positively associated in a powered partial least squares discriminant analysis along the first principal component with bees treated with dsRNA targeting fatty acid elongases (in bue: *GB51247, GB51250, GB54397, GB54404*) or fatty acid amide hydrolase (FAAH, in yellow: *GB53695*) and with control bees (in grey, injected with dsRNA targeting GFP). Asterisks indicate statistical significance of abundance differences between the two treatment groups after Holm-Bonferroni correction for multiple ($N = 5$) testing (Welch's t-test: $p \leq 0.05$ [*], $p \leq 0.01$ [**], $p \leq 0.001$ [***]).

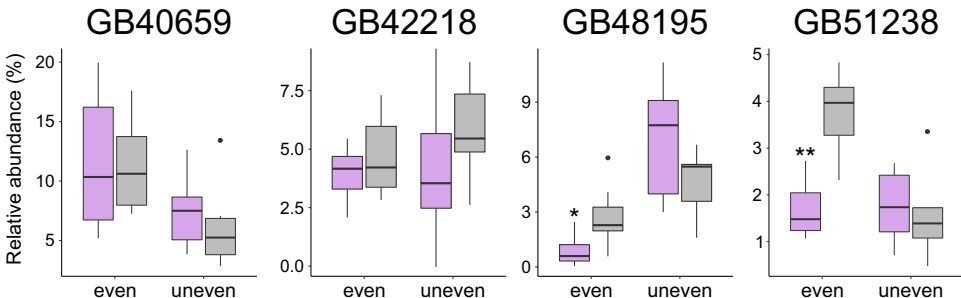

**Fig. 8 Relative abundance of alkenes with double bonds at even and at uneven positions in fatty acid desaturase RNAi-treated bees and control bees.** Box plots of the relative abundances of alkenes in bees treated with dsRNA of fatty acid desaturases (*GB40659, GB42218, GB48195, GB51238*; purple) or dsRNA of GFP (control; grey). Asterisks indicate statistically significant abundance differences between the two groups (RNAi treated vs. control) after Holm-Bonferroni correction for multiple ($N = 10$) testing (Welch's t-test/Wilcoxon signed-rank test: $p \leq 0.05$ [*], $p \leq 0.01$ [**]).

in statistically significant CHC profile changes in nine of the experiments. In the light of these results, the remaining candidate genes not further experimentally assessed in the present investigation also appear highly promising targets for future reverse genetic and heterologous expression experiments, as done, for example, by Tupec et al.[21] on fatty acyl-CoA reductases.

Our study provided strong evidence for four fatty acid elongases (*GB51247, GB51250, GB54397, GB54404*) modulating CHC profile composition in Hymenoptera (Figs. 5 and 7). Fatty acid elongases previously identified in Diptera to be involved in CHC biosynthesis belong to clade ELO C3 (*CG30008*[19]) and clade ELO C7 (i.e., *EloF*[20], *CG9458*[19], *CG18609*[19]) (Fig. 2). Our RNAi experiments on honey bees confirmed that genes of clade ELO C7 (*GB54397*) are also involved in CHC biosynthesis of Hymenoptera. They further revealed that genes of three additional major clades of this gene family, that is ELO B1 (*GB51250*), ELO B2 (*GB51247*), and ELO B3 (*GB54404*), are involved in shaping CHC profiles in this mega-diverse insect order (Fig. 2). The numerous clades with known CHC biosynthesis-related genes in the gene family tree and of candidate genes shown in this study to be expressed in oenocytes or trophocytes (see below) (Fig. 2) within this gene family makes genes belonging to clades ELO C1–7 promising targets for future knockdown experiments. In contrast, the involvement of genes of clade ELO A in CHC biosynthesis currently appears unlikely, as the only gene of this clade functionally studied so far (i.e., *baldspot* of *D. melanogaster*) participates in spermatogenesis[22].

We did not measure a statistically significant decrease in the expression levels of RNAi-targeted elongates at the point in time when we sampled the honey bees' CHC profiles (Fig. 5). This at first glance surprising result could be due to a combination of two

factors: 1) lag-time between gene expression level changes and CHC profile changes, and 2) short-term duration of the knockdown effect on the target genes' expression levels[23]. Because the time points for sampling dsRNA-treated bees had been chosen based on when phenotypic changes became visible in a subset of samples (typically 2–5 days after dsRNA injection; Supplementary Table 10), the time points were possibly too late to still detect transient gene expression level changes. We consider the alternative explanation, namely that RNAi treatment did not have an effect on the target genes' expression levels, as unlikely, because the observed phenotypic changes were in most instances consistent with the predicted function of the target genes. Moreover, the observed increase of the expression levels of three elongase genes after RNAi treatment could be explained by a temporary overexpression of these genes after their successful prior knockdown[23].

Fatty acid elongases have been shown to catalyze the elongation of long fatty acyl-CoAs[2,10,24]. Consistent with this idea, we found knockdown of fatty acid elongase genes to result in a decrease in the relative abundance of CHCs with chain lengths > 27 carbon atoms (Figs. 5 and 7). Yet, we did not find any additional differences between the knockdown phenotypes. Lack of distinct phenotypes could be due to off-target effects (i.e., knockdown of a gene coding for one fatty acid elongase simultaneously decreases the expression of another gene coding for a fatty acid elongase). However, except for two target genes, this scenario seems unlikely, as the nucleotide sequences of the synthesized dsRNAs (provided on Zenodo: 10.5281/zenodo.7334422) differed substantially from each other. The exceptions are *GB51250* and *GB54404*, which differed only in 43% of their nucleotides from each other. An alternative explanation for the lack of distinct phenotypic effects could be that the targeted fatty

acid elongases exhibit, in stark contrast to those of mammals[24], little substrate specificity. Finally, manipulation of the elongation steps could cause a shift of substrate concentrations along the entire CHC biosynthetic pathway, making it difficult to detect substrate specificity. Additional tools have to be used in future investigations to further interrogate the specific function and the substrate specificity of the enzymes encoded by the here-identified genes (e.g., heterologous expression, as done by Pei et al.[25], or in vivo synthesis using labeled precursors). The results of our study provide the foundation for such investigations.

We identified four fatty acid desaturase genes (*GB40659*, *GB42218*, *GB48195*, *GB51238*) that modulate CHC composition in worker honey bees. Previous research on *D. melanogaster* had led to the identification of three closely related fatty acid desaturases (*Desat1*, *Desat2*, and *DesatF*) for being involved in CHC biosynthesis and introducing double bonds in fatty acyl-CoA[17,18,26]. All three desaturases belong to clade Desat A1 (Fig. 3). In the honey bee, clade Desat A1 is represented by a single gene (*GB48195*), which our knockdown experiments confirmed to be also involved in CHC biosynthesis in Hymenoptera. Intriguingly, the three additional fatty acid desaturases identified in our study to influence CHC profile composition in worker honey bees belong to two clades previously not associated with CHC biosynthesis (Fig. 3): Desat A2 (*GB42218*, *GB51238*) and Desat B (*GB40659*). This demonstrates that fatty acid desaturases from very distantly related clades participate in CHC metabolism (i.e., Desat A1, Desat A2, Desat B; possibly also Desat D, which contains the *O. spinipes* candidate gene *g14708*, which lacks an ortholog in *A. mellifera*).

Knockdown of the above-outlined four fatty acid desaturases had a strong effect in worker honey bees on both target gene expression (except *GB40659*) and on CHC profile composition (Fig. 6). Moreover, we showed that knockdown of *GB48195* and of *GB51238* resulted in a decrease of alkenes, consistent with the idea of the encoded proteins introducing double bonds in fatty acyl-CoA. While all desaturases in *D. melanogaster* known to be involved in CHC biosynthesis (*Desat1*, *Desat2*, and *DesatF*) show Δ9-specificity and thus facilitate synthesis of alkenes with double bonds at uneven positions[17,18], *GB48195* and *GB51238* are the first fatty acid desaturases identified to be involved in the production of alkenes with double bonds at even positions (8 and 10) (Fig. 8). Knockdown of the remaining two fatty acid desaturases (*GB40659*, *GB42218*) led to broad and complex compositional changes involving multiple CHCs. As discussed above, lack of distinct phenotypes could be due to off-target effects, which in our case seems unlikely, however, as the nucleotide sequences of the synthesized dsRNAs (provided on Zenodo: 10.5281/zenodo.7334422) differed between all genes except two substantially from each other. The exceptions are *GB42218* and *GB51238*, whose dsRNAs differed only in 29% of their nucleotides from each other. We consider off-target effects in this specific instance to unlikely have significantly impacted the results, as the nucleotide sequence used for antisense riboprobe (ISH) and the dsRNA probe (RNAi) caused different staining in the ISH experiments and different phenotypes in the knockdown experiments, respectively. The lack of distinct phenotypes and the difficulty to detect substrate specificity could be for the similar reasons as discussed above for fatty acid elongases.

We identified a fatty-acid amide hydrolase (FAAH, *GB53695*) influencing CHC profile composition in worker honey bees. This result is remarkable, as fatty-acid amide hydrolases had never been associated with CHC biosynthesis before. Knockdown of *GB53695*, which we found to be expressed in oenocytes, resulted in a significant increase of the relative abundance of CHCs with chain lengths > 27 carbon atoms (Fig. 7). However, we did not detect statistically significant gene expression changes (Fig. 5).

Fatty-acid amide hydrolases are characterized by their amidase signature and are — in mammals — known to degrade fatty acid amides of endogenous lipids into fatty acids by removing the amide group[27]. How the functions of fatty-acid amide hydrolases relate to the modulation of CHC profiles in Hymenoptera remains unclear at this point. We deem it possible that the studied fatty-acid amide hydrolase impacts the availability of CHC precursors, by, for instance, releasing short-chain fatty acids that are then possibly reduced to aldehydes, which could be transformed to CHCs by a cytochrome p450 decarbonylase (Fig. 1). It is likely that another honey bee gene phylogenetically closer to the candidate gene *g2290* (*GB48850*) is also involved in CHC biosynthesis (Supplementary Fig. 8). Future experiments (e.g., heterologous expression) should shed light on the role of this small gene family in CHC biosynthesis.

Knockdown of candidate genes encoding a fatty acid synthase (*GB52590*) and a fatty acyl-CoA reductase (*GB52087*) resulted in a premature death of the treated bees. A similar observation was made by other researchers when targeting genes of these two families in various insects (e.g.,[16,28–31]). We hypothesize that interfering with the expression of genes of the two gene families reduced — due to their central role in CHC biosynthesis — the overall availability of CHCs to an extend that it caused desiccation-related death of the treated bees[16,29,31]. Knockdown of fatty acid synthase could additionally have reduced the synthesis of other molecules fundamentally important for survival (e.g., cuticular lipids[29]). Despite the fact that our RNAi experiments on *GB52590* and *GB52087* remained inconclusive, alternative approaches should consider them highly promising candidate genes, as multiple lines of evidence strongly suggest both genes to be involved in CHC biosynthesis: a) ortholog relationship to *D. melanogaster* genes known to be involved in CHC biosynthesis (i.e., *GB52590*, ortholog of *FASN3*[16] fatty acid synthase which belongs to clade FAS B2; Supplementary Fig. 2; *GB52087*, ortholog of the three fatty acyl-CoA reductases *CG10097*, *CG13091*, and *CG17560*[19], all of which belong to clade FAR F; Supplementary Fig. 3), b) expression in honey bee oenocytes (Fig. 4), and c) statistically significant expression differences of their *O. spinipes* orthologs (i.e, *g3158*, ortholog of *GB52590*; *g1571*, ortholog of *GB52087*) between 12 and 38 h-old *O. spinipes* females with different chemotype (Supplementary Tables 5 and 7) and between females differing in age from each other (Table 1; Supplementary Table 9). Two other FAR-encoding candidate genes, *GB50627* (clade FAR F) and *GB49380* (clade FAR B) also seem promising candidate genes, as they are both expressed in honey bee oenocytes (Fig. 4) and their *O. spinipes* orthologs (*g2413* and *g7842*) were found to be differentially expressed between females differing in age from each other (Table 1; Supplementary Table 9).

The phylogenetic distribution of genes involved in CHC biosynthesis across Euarthropoda (i.e., Chelicerata, Myriapoda, "Crustacea" (paraphyletic), and Hexapoda) has implications for our understanding of the evolution of CHC biosynthesis. Based on the four inferred gene trees and the ISH and knockdown results on *A. mellifera*, our study shows that all major gene families (Fig. 1) and most subclades within them harboring genes known to be involved in CHC biosynthesis (i.e., FAS A, ELO C, ELO B, Desat A1, and potentially FAR B) are present in the extant major lineages of Euarthropoda. The phylogenetic relationships in the gene trees of fatty acid synthases and fatty acid elongases inferred by Finck et al.[32] are consistent with this interpretation. CHCs and oenocytes have also been reported from all major Euarthropoda lineages[33–40]. We therefore consider it reasonable to hypothesize that the capacity of CHC biosynthesis evolved before the divergence of the major lineages of Euarthropoda and, consequently, before the colonization of land. An alternative hypothesis, namely convergent exaptation of the same genes for CHC biosynthesis in different lineages

of Euarthropoda, is far less parsimonious. The only gene whose phylogenetic distribution is currently not fully consistent with a CHC origin in a species inhabiting an aquatic environment encodes a cytochrome p450 decarbonylase (CYP4G gene family) (Fig. 1) that catalyzes the final step of CHC biosynthesis[8,41] and that has — so far — only been found in Hexapoda[41,42]. As the cytochrome p450 gene family is known for its high turnover rate[38], we do not put too much weight in this single exception.

Although CHCs serve social insects as important semiochemicals (e.g., cast and nest-mate recognition), social species do not possess more complex CHC profiles than solitary ones[43]. Indeed, CHCs also serve fundamental functions in solitary species (e.g., cues and signals for species and sex recognition[2]). We found the absence of systematic differences between social and solitary species in their CHC profiles to be reflected at the genetic level: none of the taxonomic lineages represented in the gene trees stands out by a particularly rich or poor set of genes, irrespective of whether or not the lineage contains solitary or social species (Figs. 2 and 3; Supplementary Figs. 2 and 3). What we found are species-specific gene duplications in *Amyelois transitella*, *Manduca sexta*, *Nasonia vitripennis*, and *Tribolium castaneum* (Supplementary Fig. 9), possibly linked to species-specific ecology and behaviors other than sociality.

We found almost all genes whose knockdown resulted in a CHC profile change to be expressed in oenocytes — consistent with the common view that CHC metabolism occurs in oenocytes (e.g.,[8,44–46]). A major exception was GB51238, encoding a fatty acid desaturase. We found this gene exclusively and highly expressed in trophocytes (Fig. 4), another cell type of the insects' fat body. While we cannot exclude the possibility that the gene is expressed at a very low level in oenocytes, we deem the result noteworthy, as other researchers previously already suggested fat body cells other than oenocytes to be possibly involved in CHC biosynthesis[16,47]. Given the possible implications of CHC biosynthesis involving additional types of fat body cells, we recommend future studies to not exclusively focus on oenocytes when studying CHC metabolism.

RNAi experiments are difficult to conduct on *O. spinipes*, because these wasps cannot be cultured (at least so far), occurs at most known field sites in comparatively small numbers, and because rearing age-controlled specimens of this species is extremely time consuming (i.e., can only be done once per year). We therefore decided to assess a posteriori whether the possible functions of genes — inferred from our knockdown experiments on honey bees — that are differentially expressed in *O. spinipes* females with different chemotypes, can explain the specific phenotypic differences between the two chemotypes. This approach rests on the ortholog conjecture, which states that orthologous genes in different species tend to retain the same function[48]. Intriguingly, two of the five genes that we found differentially expressed between *O. spinipes* females with different chemotypes (i.e., g14712, g7616) are likely sufficient to explain the majority of the phenotypic differences between the two chemotypes. Knockdown of GB51238, a co-ortholog of the *O. spinipes* fatty acid desaturase g14712, caused a decrease in the relative abundance of alkenes with double bonds at even carbon chain positions in worker honey bees. Consistent with this result, we found g14712 to be higher expressed in *O. spinipes* females expressing chemotype 2, which is characterized by alkenes with double bonds at even carbon chain positions (i.e., 8, 10, 12, and 14) than in *O. spinipes* females expressing chemotype 1, which is characterized by alkenes with double bonds at uneven carbon chain positions (i.e., 7 and 9). Likewise, the knockdown of GB51247, an ortholog of the *O. spinipes* fatty acid elongase g7616, caused a decrease of the relative abundance of CHCs with chain lengths larger than 27 carbon atoms in worker honey bees. Consistent with this result, we found g7616 to be higher expressed in *O.*

*spinipes* females expressing chemotype 2, which is characterized by a high relative abundance of alkenes with chain lengths > 27 carbon atoms, than in *O. spinipes* females expressing chemotype 1, which is characterized by low relative abundance of alkenes with chain lengths > 27 carbon atoms (Supplementary Fig. 1). Heterologous expression experiments[21] could provide the means to test our above hypotheses and to shed light on the specific enzymatic reactions catalyzed by the remaining identified candidate genes in *O. spinipes*.

Our study demonstrates how comparative transcriptomic analyses on a non-model organism featuring unique characteristics can deliver valuable information that research on established model organisms may not be able to provide. We presented an array of genes that impact CHC profile composition in Hymenoptera. These enable future experimental manipulation of CHC profiles in this mega-diverse insect order and functional investigations on the activity and substrate specificity of the encoded proteins.

## Methods and material

**Odynerus spinipes chemotype persistency analysis.** We collected prepupae of *O. spinipes* females and males from trap nests (details in Supplementary Methods). Each female was placed in a separate observation cage (30 cm×30 cm x 30 cm, Bioform, Nürnberg, Germany) and provided with a conspecific male and absorbent paper soaked with honey and water. The presence of males was meant to simulate field conditions for the females. The cages were kept in a climate chamber (70% humidity, 23 °C during day, 18 °C during night, and with a 12/12 h day/night cycle). We sampled the cuticular hydrocarbons (CHCs) of 23 females with the aid of Solid Phase Micro-Extraction (SPME) fibers (see below) at two days, seven/eight days, and 14 days (or immediately after they died) after they had eclosed (Supplementary Table 1).

We sampled the CHCs of 18 females two to four times during their adult life in the field (Supplementary Table 2). Females were caught at their nest entrance, color-coded with water-resistant paint markers (Edding 780, 0.8 mm, Ahrensburg, Germany), CHC-sampled with the aid of SPME fibers, and released the next morning at their nest site. The sampling time coincided with the start of the nest building activity.

CHCs from living wasps were sampled by stroking a SPME fiber (Supelco, coating: polydimethylsiloxane, 100 μm, Sigma-Aldrich, Bellefonte, PA, USA) for 2 min. (field populations) or 4 min. (lab population) over the cuticle of a female's metasoma. Wasps processed in 2016 were anesthetized by exposing them for 1 min. to $CO_2$, those processed in 2017 by cooling them down for 3 min. at $-20$ °C (because we suspected $CO_2$ treatment to cause amnesia in the treated wasp, as wasps treated this way in 2016 rarely returned to their nest; see[49]).

All CHC extracts were either analyzed with a gas chromatograph coupled to a mass spectrometer (GC-MS) or with a gas chromatograph-flame ionization detector equipped with a SLB-5 MS non-polar capillary (GC-FID) applying the same temperature profile (details in Supplementary Methods). The GC-MS data were recorded and quantified with the software HP Enhanced ChemStation G1701AA (version A.03.00; Hewlett Packard, Palo Alto, California, USA). The GC-FID data were recorded and quantified with the software GCSolution (Shimadzu, Wemmel, Belgium). The chemotype of the female wasps was inferred from the presence/absence of chemotype-specific alkenes[14,15] with diagnostic Kovats indices[50].

**Odynerus spinipes draft genome.** We sequenced the nuclear genome at a base-coverage depth of ca. 86x using Illumina HiSeq2000 sequencing technology and three different types of NGS libraries: 250 bp paired-end [10.58 Gbp] (55x), 800 bp paired-end

[2.45 Gbp] (13x), and 3 kbp mate-pair [3.4 Gbp] (18x), generated from a single pool of DNA extracted from males and females (details in Supplementary Methods). To foster gene annotation, we sequenced whole-body transcriptomes of one adult male and of two adult females (one of each chemotype). DNA and RNA extraction, library preparation, and Illumina DNA sequencing were done by BGI Tech Solutions (Hong Kong, PRC).

All raw reads of genomic DNA were processed with trimmomatic[51] version 0.33 (details in Supplementary Methods). The processed reads were then used to assemble the genome with Platanus[52] version 1.2.4 using all three libraries in each of the three assembly steps (contig construction, scaffolding, and gap closing) and applying the software's default parameters. The protein-coding gene repertoire was annotated with the BRAKER ab initio gene prediction pipeline[53] version 2.1 and providing the RNA-seq data of the three above-mentioned adult wasps as intrinsic evidence (details in Supplementary Methods).

**Comparative gene expression analysis**. We sequenced metasoma transcriptomes of twelve *O. spinipes* females raised under laboratory conditions from field-collected prepupae (Supplementary Table 4): six of them (three of each chemotype) were sampled 12–38 h after they had eclosed, the other six (three of each chemotype) were sampled 48–62 h after they had eclosed. The prepupae were collected from three field sites (Supplementary Table 4).

Prior to RNA fixation, the eclosed females were kept separate from each other at 21 °C in polystyrene tubes (53 mm×100 mm, Bioform, Nuremberg, Germany) containing moistened cotton and experienced a 12/12 h day/night cycle. All samples were euthanized at the same time of the day (1 pm) to minimize a potential impact of expression differences by rhythmically expressed genes (note that expression of *Desat1* follows such a diurnal rhythm in *D. melanogaster*[54]). Wasps were euthanized by submersing them for 60–120 sec. in hexane (*n*-hexane for gas chromatography, SupraSolv, Merck, Germany). The CHC extracts were used to infer the wasps' chemotypes as described above. Each wasp's metasoma was ground with a sterile pestle in 1 mL RNAlater (QIAGEN, Hilden, Germany) and kept at −80 °C before further processing it.

RNA extraction, NGS library preparation, and 150 bp paired-end nucleotide sequencing with an Illumina NextSeq 500 (Illumina, San Diego, USA) were done by StarSEQ (Mainz, Germany) using standard protocols. We collected on average 20–44 million reads per sample. All raw reads are available from GenBank in Bioproject PRJNA609595 (Supplementary Table 4).

The RNA-seq data were analyzed with software integrated in the Galaxy Europe platform (https://usegalaxy.eu). We quality-trimmed the raw reads and removed NGS library adapters with Trim Galore version 0.4.3.0 (http://www.bioinformatics.babraham.ac.uk/projects/trim_galore/). The quality of the raw reads was assessed with FastQC version 0.67 (http://www.bioinformatics.babraham.ac.uk/projects/fastqc/). The gene model file (GTF) of the *O. spinipes* genome was used to model splice junctions with the software STAR[55] version 2.5.2b0 via mapping the transcript raw reads onto the genome assembly. We inferred a matrix of normalized read counts per gene, considering the genes in the above GTF file, with featureCounts[56] version 1.5.3.

Because our initial experimental design included only one batch of six transcriptomes (three of each chemotype) (Supplementary Table 4), we applied a two-step procedure for identifying genes differentially expressed between females with different chemotypes: we first searched with DESeq2[57] version 2.11.39 for differentially expressed genes in each of the two batches, using a False discovery rate (FDR) of 0.05 as cutoff. We then reciprocally

assessed genes that this analysis indicated to be differentially expressed between the two chemotypes in the other batch, this time applying a more conservative Holm-Bonferroni correction for multiple testing. Only genes that were significantly differentially expressed in both analyses were kept as candidate genes. We assessed with edgeR[58] version 3.28.0 in the R statistical software version 3.4.1 (http://www.R-project.org) whether the five genes identified with the above-described procedure were also found differentially expressed in the two batches (*p*-value< 0.05 after Holm-Bonferroni correction for five tests). We used DESeq2 as well as edgeR to assess the statistical significance of gene expression differences between *O. spinipes* females of different age classes, ignoring the wasps' chemotype. We considered genes, judged to differ in their expression by either of the two programs, as candidates if they encoded fatty acid synthases (FAS), fatty acid elongases (ELO), fatty acid desaturases (DESAT), or fatty acyl-CoA reductases (FAR).

**Identification of candidate gene orthologs in the honey bee genome**. Honey bee orthologs of *O. spinipes* candidate genes belonging to the gene families Desat, ELO, FAR, and FAS were identified by their phylogenetic position in the corresponding gene trees. To this end, we identified the proteins of the four gene families by searching for proteins with the protein family-characteristic protein domains (see below) available as HMM models from PFAM database[59] (PFAM-A, version 28). The HMM models were searched with the aid of hmmer[60] version 3.1b1 against the protein sets of 37 Euarthropoda, including *O. spinipes*, with sequenced genomes (details in Supplementary Methods). To identify fatty acid synthases, we searched for proteins containing the PFAM domains ketoacyl-synt KS (Pfam family accession PF00109), acyl-transferase-1 (PF00698), and PS-DH (PF14765) (e-value < 1e-5). To identify fatty acid elongases, we searched for proteins containing the ELO PFAM domain PF01151 (e-value < 1e-30), and consisting at least of 200 amino acids. To identify fatty acid desaturases, we searched for proteins containing the PFAM domain PF00487 (e-value < 1e-20). Note that we did not consider sphingolipid desaturases, as they show little amino acid sequence similarity to other desaturases (see[12]). To identify fatty acyl-CoA reductases, we searched for the simultaneous presence of the two PFAM domains PF07993 (e-value < 1e-57) and PF03015 (e-value < 1e-15). To identify fatty acid amide hydrolases, we searched for proteins containing the amidase PFAM domain PF01425 (e-value < 1e-65). The amino acid sequences of a given gene family were aligned with MAFFT[61] version 7.123 and phylogenetically analyzed with the software IQ-TREE[62] version 1.6 (details in Supplementary Methods).

Honey bee orthologs of the remaining *O. spinipes* candidate genes and which did not belong to the above gene families were identified by applying the best reciprocal hit criterion and searching the protein repertoires of the two species with the blastp program of the BLAST + software suite[63,64].

**In situ hybridization**. RNA probes were synthesized according to standard protocols[65] from cloned partial mRNA sequences of all candidate genes (Tables 1 and 2) using DIG and FLU RNA Labeling Kits (Roche, Basel, Switzerland). Strand cDNA was synthesized from isolated total RNA (1 μg, DNase-treated) of fat body tissues isolated from *A. mellifera* using the Quantitect kit (QIAGEN GmbH, Hilden, Germany). The resulting cDNA library was used to amplify target regions via polymerase chain reactions (PCRs) (Mastercycle DNA Engine Thermal cycler PCR, Eppendorf AG, Hamburg, Germany) using the Phusion high fidelity kit (New England Biolabs, Ipswich, MA, USA). Amplicons were cloned with the NEB PCR Cloning kit (New England Biolabs GmbH, Ipswich, MA, USA), and the plasmids were extracted

using the Gen Elute Plasmid Miniprep kit (Sigma-Aldrich, Steinheim, Germany) and were sent to Genewiz (Genewiz, Leipzig, Germany) for nucleotide sequencing. The cloned gene fragments (387–774 bp long) were used to synthesize anti-sense and sense RNA probes using T7 (Roche, Mannheim, Germany) or SP6 (Roche, Mannheim, Germany) RNA polymerases. Sense probes served as negative control to each anti-sense probe. Oligonucleotide primers used for probe cloning were designed with the online software Primer-BLAST[66] and synthesized by Metabion (Planegg, Germany) or Sigma-Aldrich (Steinheim am Albuch, Germany) (Supplementary Table 11).

In situ hybridization was done on worker honey bees collected from a single hive at Würzburg University. All bees eclosed 4–8 days prior to their processing, were reared from prepupae in climate chambers at 28 °C, and were collectively kept in observation cages at 32 °C with 30% sugar solution. Metasoma tergites and sternites were dissected from living bees (placed on ice for 5 min) in 4% paraformaldehyde (PFA) (Sigma-Aldrich, Steinheim am Albuch, Germany; dilution in Phosphate buffered saline [PBS]). The dissected segments were fixed overnight in a solution consisting of 2 mL heptane (Carl Roth, Karlsruhe, Germany) and 2 mL of 4% PFA on a rotator at room temperature. The fixation was finalized following the protocol given by Dearden et al.[67]. Whole-mount RNA in situ hybridization was done as described by Dearden et al.[68], with minor modifications outlined in Supplementary Methods. All mounting media included DAPI (0.5 μg/mL; Thermofisher Scientific, Waltham, Massachusetts, USA) to achieve nuclear counterstaining. High-resolution images of stained tissues mounted on glass slides were obtained with a Zeiss Axio Imager A1 fluorescence microscope, a Zeiss AxioCam MRc, DAPI fluorescence filters, a Zeiss HXP120 UV light unit, and the Zeiss Axiovision software (Carl Zeiss Jena GmbH, Jena, Germany). Overlay of brightfield images and DAPI fluorescence was done with ImageJ/Fiji (https://fiji.sc/). Trophocytes were distinguished from oenocytes by their polyploid or polygonal cells (oenocytes are round) with a large irregular nucleus (the nucleus of oenocytes is round or oval)[69].

**RNAi-mediated knockdown experiments**. Double-stranded RNA (dsRNA) was synthesized in vitro from 387–774-bp-long inserts — previously generated for RNA probe synthesis — using MEGAscript RNAi kit (Ambion, Texas, USA) (details in Supplementary Methods). T7 promotor was added to the oligonucleotide primers (listed in Supplementary Table 11) used to amplify the plasmids. A plasmid with a DNA fragment encoding green fluorescent protein (GFP), kindly provided by Anne-Kathrin Rohlfing (nucleotide sequence on Zenodo: 10.5281/zenodo.7334422), was used to synthesize dsRNA that served as negative control. We verified that the GFP nucleotide sequence had no significant match in the honey bee genome[70] by searching it with tools of the BLAST + software suite[63,64] against the honey bee genome (GCF_000002195.4_Amel_4.5). The identity of the amplicons was verified by nucleotide-sequencing them. All dsRNA probes were diluted in MOPS Ringer solution (270 mM NaCl, 3.2 mM KCl, 1.2 mM CaCl₂, 10 mM MgCl2, 10 mM γ-(N-Morpholino) propane sulfonic acid (MOPS), ph 7.4) to reach a final concentration of 5 μg/μL.

Knockdown experiments were done on worker honey bees reared as described above and chilled prior to the dsRNA injection. All bees of a given experiment came from the same hive, eclosed 6–7 days prior to dsRNA treatment and belonged to the same caste (workers, nurse bee). We injected between the 2ⁿᵈ and 3ʳᵈ tergite of each bee's the metasoma 1 μL dsRNA (5 μg) with a Hamilton microliter syringe (MICROLITER Series 7000 Tip type 2, 7105 KH, Carl Roth, Karlsruhe, Germany). Per the

experiment, we treated 6–12 bees with dsRNA of the target gene of interest and 6–13 bees with dsRNA of GFP, which served as control (Supplementary Table S11). Bees of different experiments were kept in separated observation cages and were fed with fed 30% sugar water. Note that we were forced to use the same control group when assessing knockdown of GB51250 and GB48195 (which have been carried out at the same time and with bees from the same hive), as we lacked bees to make an extra control group at that specific time.

After dsRNA injection, we daily sampled the CHCs of six bees (three of the treatment group and three of the control group) from a given RNAi experiment with SPME fibers, probing each bee's metasoma. As soon as notable CHC profile differences between treatment bees and control bees were detected (sampling times given in Supplementary Table 10), all samples from the corresponding RNAi experiment were euthanized by immersing them for 2 min. in hexane (n-hexane for gas chromatography, SupraSolv, Merck, Germany) and were subsequently frozen in liquid nitrogen. The bees' metasoma were subsequently stored at −80 °C before further processing them.

CHC extracts were analyzed with a GC-MS (details in Supplementary Methods). CHC abundance values were normalized by dividing them by the total abundance of CHCs identified in a given CHC extract (available on Zenodo: https://doi.org/10.5281/zenodo.7334422). The normalized data were analyzed in Rstudio (R version 4.0.3), using functionalities from the following packages: RVAideMemoire version 0.9–80 (https://CRAN.R-project.org/package=RVAideMemoire), vegan version 2.5–7, Hotelling version 1.0–7, pls version 2.7–3, FactoMineR[71] version 2.4, and Factoextra version 1.0.6 (https://CRAN.R-project.org/package=factoextra). The statistical significance of CHC profile differences between groups was assessed with a powered partial least squares discriminant analysis (PPLS-DA) and applying pairwise permutation tests[72,73]. In order to test the effect of RNAi of elongases on CHC profiles, we plotted the chain lengths of cuticular hydrocarbons (CHCs) positively associated (threshold = 0.5, CHC plots available on Zenodo: https://doi.org/10.5281/zenodo.7334422) in a PPLS-DA with the first principal component (Fig. 5: correlation circle plots) separating bees treated with dsRNA targeting fatty acid elongases (GB51247, GB51250, GB54397, GB54404) or a fatty acid hydrolase (GB53695) and with bees treated with dsRNA targeting GFP (control).

Differences in chain lengths and the relative abundance of specific alkenes were assessed with Welch's t-test or with the Wilcoxon signed-rank test, depending on whether or not the data appeared normally distributed (assessed with the Shapiro-Wilk test), using the stats package version 3.6.2 in Rstudio. P-values were corrected with the Holm-Bonferroni method for multiple testing. Boxplots were drawn with the ggplot2 package (https://ggplot2.tidyverse.org) in Rstudio.

We extracted the total RNA from the metasoma of each bee treated with dsRNA using RNeasy Plus Mini kit (QIAGEN, Hilden, Germany) and quantified the relative expression of the target genes via real-time quantitative PCR (RT-qPCR) using the QuantiTect SYBR Green RT-PCR kit (QIAGEN, Hilden, Germany) and a StepOnePlus Real-Time PCR System (Applied Biosystems-Life technologies, ThermoFisher Scientific, Waltham, Massachusetts, USA). We quantified expression levels in at least five samples per group (i.e., treatment and control) and in three technical replicates per sample. RT-qPCR oligonucleotide primers used to assess the target genes' expression levels are listed in Supplementary Table 12 and were designed with Primer-BLAST[66]. If possible, oligonucleotides were designed so that the resulting amplicon was 150–250 bp in length, that the oligonucleotides' melting temperature was between 57 and 63 °C, that the oligonucleotides would span exon-exon junctions, that the amplified region was localized at the 5' to

the dsRNA target region[74], and that the amplified region was not overlapping with the dsRNA target region. We used *GB49975* (predicted ribosomal protein L1) as reference gene (Supplementary Table 11), because its *O. spinipes* ortholog was one of the genes that exhibited the least expression differences between females with different chemotype. All oligonucleotide primer pairs exhibited an amplification efficiency of more than 95%, and each amplicon showed a single melting temperature maximum. The expression data were analyzed following the protocol given by Pfaffl, 2001[75]. Gene expression levels were statistically assessed with the Welch's t-test and with the Wilcoxon signed-rank test, as described above. *P*-values were corrected with the Benjamini-Hochberg method for multiple testing.

**Reporting summary**. Further information on research design is available in the Nature Portfolio Reporting Summary linked to this article.

## Data availability

The nucleotide sequence data of the genome shotgun project are deposited under the NCBI Bioproject PRJNA735081. The draft genome assembly and the corresponding gene annotations are available under the NCBI accession JAIFRP000000000 and from Zenodo (https://doi.org/10.5281/zenodo.7334422). The transcriptomic raw reads are available from GenBank in Bioproject PRJNA609595. Additional data that support the findings of this study have been deposited on Zenodo (https://doi.org/10.5281/zenodo.7334422).

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

## Acknowledgements
We are indebted to Karl-Heinz Schmalz and Wolf-Harald Liebig for providing us *O. spinipes* samples and to Jean-Yves Baugnée for providing information where to find *O. spinipes* in Belgium. We thank Michael Schnell, Harald Noeske, and Rainer Blum for help with building trap nests. We are thankful to the following people for fruitful discussions, help in the lab, and/or with handling bees and wasps: Rabea Blümel, Dr. Nicolas Braseroa, Dr. Ruth Castillo, Dr. Laura Degirmenci, Janina Diehl, Claudia Etzbauer, Dr. Hermes Escalona, Martin Gabel, Dr. Ewald Grosse-Wilde, Wolf Haberer, Sinan Kaya-Zeeb, Sabine Knaup, Sandra Kukowka, Ruth Lieberth, Dr. Jana Maehner, Dr. Baptiste Martinet, Dr. Christine Missbach, Karin Möller, Maya Nakajima, Dr. Fabian Rüdenauer, Florentine Schaub, Felix Schilcher, Doris Waffler, Katharina Wagler. We are thankful to Mareen Geyer for help with analyzing CHC profile data. We are grateful to Dr. Daniel Elsner, Dr. Jose Manuel Monroy Kuhn, and Dr. Karen Meusemann for their support regarding differential gene expression analyses. We thank Prof. Dr. Judith Korb and PD Dr. Volker Nehring for advice regarding statistical analyses, and we thank Dr. Thomas Pauli for bioinformatic support. We acknowledge support by the Freiburg Galaxy Team: (i.e., Prof. Rolf Backofen, Bérénice Batut, Bjoern Gruening, Torsten Houwaart), funded by the Collaborative Research Centre 992 Medical Epigenetics (DFG grant SFB 992/1 2012) and German Federal Ministry of Education and Research (BMBF grant 031 A538A RBC). We acknowledge the Struktur- und Genehmigungsbehörde Süd and the Struktur- und Genehmigungsdirektion Nord (both Rhineland Palatinate, Germany) for granting permission to collect samples and Gaby Schöning for granting permission to place trap nests at a bee hotel in Büchelberg. Major parts of the present study were funded by the German Research Foundation (DFG; NI1387/2-1, SCHM 2645/6-1).

## Author contributions
O.N., T.S., and V.C.M. conceived and designed the project. A.D., B.M., J.W., M.P., O.N., and S.M. sequenced, assembled, and annotated the *Odynerus spinipes* genome. J.P.O. and V.C.M. performed the comparative transcriptomic analyses. L.P. and V.C.M. inferred the gene trees. D.L. and V.C.M. designed the in situ hybridization experiments. M.T., O.N., R.S., T.S., and V.C.M. contributed to the design of the knockdown experiments. V.C.M. conducted the chemotype persistency, the in situ hybridization, the knockdown experiments, and all chemical and statistical analyses. D.L., O.N., R.S., T.S. provided instruments and reagents. O.N. and V.C.M. wrote the manuscript, with contributions from T.S. All authors provided feedback on the manuscript.

## Funding

## Competing interests
The authors declare no competing interests.
