## [Peer Review File · Communications Biology]

Reviewers' comments:

Reviewer #1 (Remarks to the Author):

Brief Summary:

The manuscript presented by Moris et al. describes a series of thoughtful investigations into the genetic basis of cuticular hydrocarbon (CHC) production with a focus on the insect order Hymenoptera. The study combines experimental work on a mason wasp and the honey bee to identify and functionally characterize genes contributing to CHC biosynthesis. Given the widespread orthology of these genes across arthropods, the authors make the case that the production of CHCs was a trait exhibited in the aquatic common ancestor of arthropods.

General impressions:

My overall impression of this work is very positive. Moris et al. have contributed interesting new results that advance our understanding of the biosynthesis of CHCs in insects, especially Hymenoptera.

The use of solid-phase microextraction to non-destructively re-sample the cuticular hydrocarbons of live mason wasps, for the purpose of confirming the qualitative stability of the chemotypes, is a nice approach. Moreover this re-sampling sheds light on the developmental shift in quantitative CHC composition which justifies measuring differential gene expression over time.

Moris et al. insightfully leverage the intrasexual cuticular chemical polymorphism in the mason wasp *Odynerus spinipes* and the developmental shifts in CHC composition in the same to identify genes differentially expressed (1) between chemotypes and (2) between developmental time periods. Given that most of these genes have orthologs in the honey bee, Moris et al. make the reasonable decision to carry out further molecular investigations in the honey bee, which is a more established "model system" than the mason wasp. It would be exciting to see these molecular investigations carried out in the mason wasp, but this is presumably experimentally prohibitive in ways that perhaps could be discussed a bit more, to assure a more skeptical reader that the decision to focus on the honey bee for functional work is justified.

Another line of evidence that would support the claim that the CHC production genes have conserved specific functions in the honey bee and the mason wasp might be coding sequence conservation (e.g., percent amino acid identity between orthologous copies in mason wasp vs. honey bee), but the presence of the family-characteristic protein domains is sufficient to support the claim that the orthologs have shared general functions in desaturating, elongating, etc.

The use of in-situ hybridization and RNA interference to localize and alter the expression of a variety of CHC production genes in the honey bee strikes me as novel. In addition to the localization of CHC production gene expression in specific cell types in the honey bee abdomen, the in-situ hybridization work is informative to anyone interested in fat body cell type diversity in the honey bee and in Hymenoptera more generally. The use of sense probes as a control verifies that the in-situ protocol was reliable. The use of dsRNA targeting jellyfish green fluorescent protein as a control for dsRNA injection helps to verify that the effects of target gene dsRNAs are the result of their specific sequences matching genes of interest, and not an artifact of dsRNA injection.

I think this study will be of interest to a range of workers across subdisciplines. The study is well conceived and the results reported are compelling, so the manuscript is in great shape as it is. My comments are suggestions that I hope might help to make the manuscript as clear as possible.

Specific minor comments:

[1] Line 55: It seems like many genes corresponding to the enzymes involved in CHC biosynthesis have been identified, which is why a candidate gene approach is possible (e.g., Finck J et al. 2007. Divergence of cuticular hydrocarbons in two sympatric grasshopper species and the evolution of fatty acid synthases and elongases across insects. *Scientific Reports* 6:33695). Candidate genes have been identified based on our understanding of the enzymatic steps of CHC biosynthesis, but their function in shaping the CHC profile is not clear. This is the gap that your laudable work helps

to fill.

[2] Line 75: "eusociality" is the term more commonly used, but "eusocialism" does appear a handful of times in the literature. My personal preference would be to use the more commonly used term, but if you prefer the other term that is fine too.

[3] Line 140: please double check the N50 value. My impression is that the N50 is 3.637 Mbp.

[4] Line 145, 147: please double check NCBI Bioproject number and NCBI accession number

[5] Line 154: Readers may find it easier to understand the comparison made if you were to add the phrase "between chemotypes" e.g., "Considering only genes that were consistently differentially expressed between chemotypes in both batches ..." Table 1 helps to clarify the exact comparisons made. At Line 157: For clarity, it might help to change "in the two batches" to "between the two batches"

[6] Conclusions drawn in the Discussion section logically flow from the presented results. Moris et al. do a good job of presenting alternative explanations for the patterns of gene expression observed.

[7] Line 299: The discussion considers reasons why a decrease in the expression of ELO RNA was not observed at the time of sampling. It might be worth offering a sentence or two addressing the positively trending expression levels of 3 of the ELO genes after RNAi treatment. Hypothetically, autoregulation of ELO gene expression might explain this trend. In this scenario, reduction of elongase proteins would result in the observed phenotypic shift while also resulting in the overcompensation of ELO transcription and higher ELO transcript abundance. That is if ELO proteins contribute to negatively regulating transcription of ELO, absence of ELO proteins (e.g., after dsRNA treatment) would result in higher ELO expression as a result of less inhibited transcription of ELO.

[8] Line 309: minor grammatical error: "...because of the observed phenotypic changes were in..." consider deleting "of"

[9] Line 368: Here off-target effects are discussed as a possible explanation for the lack of distinct phenotypes resulting from DESAT RNAi knockdown. The above section on ELO (326-337) is more comprehensive in considering also substrate specificity and the possible shift in substrate concentrations along the entire pathway. It may be worth while to briefly mention these alternative explanations for DESAT too.

[10] Line 446: It is interesting to know precisely how chemotype 1 and chemotype 2 differ chemically, and it may be helpful to the reader to be informed about this in the introduction, so that they can have it in mind that alkenes are particularly variable between chemotypes when reading the results section. Around line 99 it may be worth while to include a bit about the previously published finding that alkenes especially differ between chemotypes.

[11] Line 1042: If presenting a numbered list in the Table 1 legend, it may be best to provide a list with as many items as there are columns in the table, to avoid confusion. For example, in Table 2, there is a list of four items in the legend, corresponding to a table with four columns. But for Table 1 the list has four items and the table has five columns. Easily solved by adding "Wasp gene ID" as list item number 1 in the Table 1 legend.

[12] Line 1078: The vertical scale bar, representing number of gene copies in collapsed clades, in Figure 2 may need to be longer. For example, the vertical height of the fan edge containing three Dipteran ELO C7 genes should, according to the scale bar, containing about 10 genes, if I am interpreting correctly. Line 1083: Please double check the scale bars here too.

[13] Line 1099: Figure 5 legend, please include meaning of color code in PPLS-DA score plots

[14] Line 1105, 1122: Figure 5 and 6 legends, minor typos: please change "correction" to

"correlation" circle plot

[15] Line 1107: Figure 5 legend, please include meaning of color code in box plot

[16] Supplementary PDF Line 90: searched the protein sets, not the genomes, is that right?

[17] Supplementary PDF Line 423: minor typo: please change "correction" to "correlation" circle plot

Reviewer #2 (Remarks to the Author):

This manuscript by Moris et al. focuses on understanding CHC synthesis in two hymenopteran species: the mason wasp, *Odynerus spinipes*, and the honeybee, *Apis mellifera*. Based on their experiments, the authors conclude in their abstract "Our results set the base for experimental CHC profile manipulation in Hymenoptera and imply that the evolutionary origin of CHC biosynthesis predates the arthropods' colonization of land." I agree with the authors that the data presented in this paper contributed to experimental CHC profile manipulation in Hymenoptera. Even though as I agree with the authors' claim that the evolutionary origin of CHC biosynthesis likely predates the arthropods' colonization of land, this manuscript does not provide additional data beyond what is already known to support this. This manuscript contains a lot of experiments and analyses. However, after reading this manuscript, I feel that this manuscript feels like at least two different manuscripts pieced together without a coherent message.

Major comments

1/ In this manuscript, the authors sequenced the genome of *Odynerus spinipes* and use transcriptomics to investigate which CHC synthesis genes may be involved in the two different CHC chemotypes in this species. They then perform in situ hybridization of CHC genes in *Apis mellifera* and performed RNAi knockdown of these genes in the honeybee. The authors then try to correlate experimental data in the honeybee to possible functions on these orthologous genes in the mason wasp. To me, this sounds like two different sets of experiments for two different manuscripts. While I have no problem with the authors doing this, based on the current set of experiments, some of the conclusions cannot be supported. CHC genes evolve rapidly across species, and this has been shown even within a single genus, *Drosophila*, as the elegant work of Claude Wicker-Thomas and colleagues have showed in the past few decades. The CHCs genes expressed in *Apis mellifera* may not be expressed in similar cells as in *Odynerus spinipes*. This has been shown by Wicker-Thomas and colleagues that the expression genes like *desatF* and *eloF* evolved rapidly even within closely related species. In addition, there could be protein coding changes that changes substrate specificity of orthologues between species, and this is not taken into account. Therefore, data on gene expression and RNAi knockdown in *Apis* cannot be conclusively transferred to *Odynerus*. Experiments that would be convincing would be to knockdown these genes in *Odynerus* or to express *Odynerus* genes in a cell culture system, similar to the Tupec et al 2019 *Elife* paper, which the authors failed to cite. This is perhaps the conceptual flaw in this manuscript.

2/The discovery that a fatty acid amide hydrolase affecting CHC profile composition is novel and promising but is buried by this manuscript's many different and diverse conclusions.

3/The authors hypothesized that the "capacity of CHC biosynthesis evolved before the divergence of the major lineages of Euarthropoda" is based on i) "genes known to be involved in CHC biosynthesis (*FAS A*, *ELO C*, *ELO B*, *Desat A1*, and potentially *FAR C*) are present in the extant major lineages of Euarthropoda" and that ii) "CHCs and oenocytes have also been reported from all major Euarthropoda lineages". I agree with the authors, but the alternative hypothesis is that these genes are also involved in fatty acid synthesis for other endogenous functions and are present for that function. Oenocytes are also present in many species not just for CHC production but for other endogenous functions (Gutierrez et al. *Nature* 2007, Makki et al. *Annual Review of Ent*

2014).

Reviewer #3 (Remarks to the Author):

Cuticular hydrocarbons are important chemical signals in insects mediating various interactions among individuals. This manuscript investigates the genome and transcriptome of mason wasp to identify the candidate genes that mediate CHC dimorphism in this species. RNAi-based knockdown of the candidate gene orthologs was applied on honeybees, and their CHC profiles were compared with control individuals. Although the effects of RNAi on target gene expression is controversial, most candidate gene modulates the CHC profiles as expected. Localization of candidate genes was also investigated, and some genes were expressed in trophocytes of the fat body. Generally, the study is interesting and contributes to our understanding of molecular bases of CHC diversity in insects, however as outlined in the comments below, the results of the manuscript are descriptive and the objective and conclusion of the study are obscure.

1.

The communications via CHCs have important roles in social Hymenoptera, and identifying genes involved in CHC biosynthesis of Hymenoptera is, indeed, of major interest (Line 74-80).

In this study, the authors identified several elongase and desaturase as genes involved in CHC profiles. The product of these gene families is well-known as biosynthetic enzymes for CHCs and several pheromones in non-Hymenopteran insects. In both families, multiple genes from multiple clades were involved in CHC biosynthesis. Are there any characteristics in each clade, such as the Hymenoptera-specific diversification pattern? The authors discussed the evolution of CHC biosynthesis genes Euarthropoda level, but the evolution of gene repertoire at more local levels (i.e. family levels) will be needed for this manuscript.

In addition to these genes, a fatty-acid amide hydrolase is a novel CHC biosynthesis gene identified in this study. Does this gene have any important function for CHC signaling in Hymenoptera? How diversified is this gene family in insects? Despite its importance and novelty, the analysis of this gene is insufficient.

Overall, although the importance of Hymenopteran CHCs is emphasized in the Introduction, much of the discussion is limited to functional descriptions of individual genes in insects.

2.

The molecular bases of two distinct CHC phenotypes in *O. spinipes* is another (sub) topic of this study. It is necessary to explain more about previous studies of this species. Although the mitochondrial haplotypes have not differed between the two phenotypes (Wurdack et al., 2015 Proc. R. Soc. B 282 20151777), it is still possible that the two phenotypes are genetically determined not in the transcript levels. Even if *O. spinipes* females with different chemotypes do not appear to differ in any other trait from each other (Line 102-103), the expression of the CHC biosynthesis gene is not the only possible explanation. Are genome sequencing, assembly, and annotation of mixed male and female DNA valid, even when the phenotype is genetically determined? Please explain and discuss the possible mechanisms of CHC phenotypes in this species.

I suggest that this manuscript is better presented as two separate papers so that the two topics (one for wasps and the other for honeybees) can be discussed clearly and in detail.

Minor comments:

Line 239-241: Please show the results of GB51236 as the figure, at least in supplements.

Figure 5-6: In score plots, both axes indicate the principal components calculated by PCA? If not (i.e. it is a score of DA), a simple "component 1" or "score 1" will be better to avoid confusion.

Figure 7: I couldn't understand why the asterisk indicates the significant abundant differences. Y-axis is indicated as chain length. Then, why the chain lengths are so different among the control of each gene treatment? Is that mean the injection of GFP dsRNA also modulate the chain length of CHCs?

Reviewer 1:

Brief Summary:

The manuscript presented by Moris et al. describes a series of thoughtful investigations into the genetic basis of cuticular hydrocarbon (CHC) production with a focus on the insect order Hymenoptera. The study combines experimental work on a mason wasp and the honey bee to identify and functionally characterize genes contributing to CHC biosynthesis. Given the widespread orthology of these genes across arthropods, the authors make the case that the production of CHCs was a trait exhibited in the aquatic common ancestor of arthropods.

General impressions:

My overall impression of this work is very positive. Moris et al. have contributed interesting new results that advance our understanding of the biosynthesis of CHCs in insects, especially Hymenoptera.

The use of solid-phase microextraction to non-destructively re-sample the cuticular hydrocarbons of live mason wasps, for the purpose of confirming the qualitative stability of the chemotypes, is a nice approach. Moreover this re-sampling sheds light on the developmental shift in quantitative CHC composition which justifies measuring differential gene expression over time.

Moris et al. insightfully leverage the intrasexual cuticular chemical polymorphism in the mason wasp *Odynerus spinipes* and the developmental shifts in CHC composition in the same to identify genes differentially expressed (1) between chemotypes and (2) between developmental time periods. Given that most of these genes have orthologs in the honey bee, Moris et al. make the reasonable decision to carry out further molecular investigations in the honey bee, which is a more established "model system" than the mason wasp. It would be exciting to see these molecular investigations carried out in the mason wasp, but this is presumably experimentally prohibitive in ways that perhaps could be discussed a bit more, to assure a more skeptical reader that the decision to focus on the honey bee for functional work is justified.

Thank you for your thorough analysis of our manuscript. We appreciate your suggestion to provide more details of why we conducted the RNAi experiments on *Apis mellifera* instead of *Odynerus spinipes*. We added sentences in the Introduction and in the Discussion to make it better comprehensible why we conducted the RNAi experiments the honey bee and why comparable experiments would indeed be difficult conduct in the mason wasp:

lines 122–124:

“and testing the functional involvement of the candidate genes in CHC biosynthesis in the honey bee — in a more trackable species of Hymenoptera for reverse genetic analyses than the mason wasp”

lines 129–132:

“We then identified candidate gene orthologs in *Apis mellifera* genome to facilitate reverse genetic analyses in species that allows us to study a larger number of replicates per knockdown experiment than *O. spinipes*.”

We also added one sentence in the discussion (also asked by reviewer 2, lines 520–524):

“RNAi experiments are difficult to conduct on *O. spinipes*, because *O. spinipes* cannot be cultured (at least so far), occurs at most known field sites in comparatively small numbers, and because rearing age-controlled specimens of this species is extremely time consuming (i.e., can only be done once per year).”

Another line of evidence that would support the claim that the CHC production genes have conserved specific functions in the honey bee and the mason wasp might be coding sequence conservation (e.g., percent amino acid identity between orthologous copies in mason wasp vs. honey bee), but the presence of the family-characteristic protein domains is sufficient to support the claim that the orthologs have shared general functions in desaturating, elongating, etc.

Thanks for the suggestion to provide this additional information. We added it in column 4 of Table 1.

The use of in-situ hybridization and RNA interference to localize and alter the expression of a variety of CHC production genes in the honey bee strikes me as novel. In addition to the localization of CHC production gene expression in specific cell types in the honey bee abdomen, the in-situ hybridization work is informative to anyone interested in fat body cell type diversity in the honey bee and in Hymenoptera more generally. The use of sense probes as a control verifies that the in-situ protocol was reliable. The use of dsRNA targeting jellyfish green fluorescent protein as a control for dsRNA injection helps to verify that the effects of target gene dsRNAs are the result of their specific sequences matching genes of interest, and not an artifact of dsRNA injection.

I think this study will be of interest to a range of workers across subdisciplines. The study is well conceived and the results reported are compelling, so the manuscript is in great shape as it is. My comments are suggestions that I hope might help to make the manuscript as clear as possible.

We are really grateful for the deep analysis of our paper made by reviewer 1 as well as for his/her comments. We considered all of them and modified the text accordingly.

Specific minor comments:

[1] Line 55: It seems like many genes corresponding to the enzymes involved in CHC biosynthesis have been identified, which is why a candidate gene approach is possible (e.g., Finck J et al. 2007. Divergence of cuticular hydrocarbons in two sympatric grasshopper species and the evolution of fatty acid synthases and elongases across insects. *Scientific Reports* 6:33695). Candidate genes have been identified based on our understanding of the enzymatic steps of CHC biosynthesis, but their function in shaping the CHC profile is not clear. This is the gap that your laudable work helps to fill.

Thank you for this comment. If it is indeed true that the main enzymatic steps have been identified with labeled carbons and some genes have been identified to perform these steps. These genes have been identified only in a few species, such as *Drosophila*. Only a single gene has been verified to be functionally used for CHC biosynthesis by Hymenoptera (a p450 cytochrome decarbonylase). Therefore, the primary aim of this study was to identify genes involved in CHC biosynthesis in a Hymenoptera. Indeed, CHC biosynthesis in Hymenoptera might differ from the one described for Diptera. We expected finding largely orthologs of genes previously

reported involved in CHC biosynthesis based on research on, for example, *Drosophila*, but our approach also allowed us to identify new candidate genes, coding for proteins other than the ones known so far. This is how we found that a FAAH impacts CHC profile composition in *A. mellifera*. How the main enzymes (already described) shape CHC profile was only a secondary objective of our study. We give some indications in the discussion the presence of which CHCs these enzymes impact. For this reason, and in order to keep the abstract short (max. 150 words), we kept the original sentence. However, we added one sentence at the end of the introduction to introduce this idea (lines 136-139):

“These experiments allowed us to identify an array of genes impacting the honey bee’s CHC profile composition and provided hints what functions the encoded enzymes might have in the CHC biosynthesis of this species and of Hymenoptera in general.”

[2] Line 75: "eusociality" is the term more commonly used, but "eusocialism" does appear a handful of times in the literature. My personal preference would be to use the more commonly used term, but if you prefer the other term that is fine too.

It is a good idea indeed to choose the most commonly used term, we replaced eusocialism by eusociality.

[3] Line 140: please double check the N50 value. My impression is that the N50 is 3.637 Mbp.

Thank you very much for point this conversion error out to us: we corrected the N50 value!

[4] Line 145, 147: please double check NCBI Bioproject number and NCBI accession number

It is not released yet, it will be as soon as our manuscript is published. If you wish to see the files that we uploaded on NCBI, you can access them via this link:

<https://oc.zfmk.de/index.php/s/kLMmqyPHrDYE94q>

[5] Line 154: Readers may find it easier to understand the comparison made if you were to add the phrase "between chemotypes" e.g., "Considering only genes that were consistently differentially expressed between chemotypes in both batches ..." Table 1 helps to clarify the exact comparisons made. At Line 157: For clarity, it might help to change "in the two batches" to "between the two batches"

We added “between chemotypes” and “between the two above batches” as suggested. It indeed makes it clearer which comparisons we did.

[6] Conclusions drawn in the Discussion section logically flow from the presented results. Moris et al. do a good job of presenting alternative explanations for the patterns of gene expression observed.

Thank you for this nice remark.

[7] Line 299: The discussion considers reasons why a decrease in the expression of ELO RNA was not observed at the time of sampling. It might be worth offering a sentence or two addressing the positively trending expression levels of 3 of the ELO genes after RNAi

treatment. Hypothetically, autoregulation of ELO gene expression might explain this trend. In this scenario, reduction of elongase proteins would result in the observed phenotypic shift while also resulting in the overcompensation of ELO transcription and higher ELO transcript abundance. That is if ELO proteins contribute to negatively regulating transcription of ELO, absence of ELO proteins (e.g., after dsRNA treatment) would result in higher ELO expression as a result of less inhibited transcription of ELO.

We are grateful for the suggestion. Since you suggested it as well, we added one sentence to discuss it (lines 355–358):

“Moreover, the observed increase of the expression levels of three elongase genes after RNAi treatment could be explained by a temporary overexpression of these genes after their successful prior knockdown²³.

23: Sapountzis, P., Duport, G., Balmand, S., Gaget, K., Jaubert-Possamai, S., Febvay, G., et al. (2014). New insight into the RNA interference response against cathepsin-L gene in the pea aphid, *Acyrtosiphon pisum*: molting or gut phenotypes specifically induced by injection or feeding treatments. *Insect Biochem. Mol.* 51, 20–32. doi: 10.1016/j.ibmb.2014.05.005

[8] Line 309: minor grammatical error: "...because of the observed phenotypic changes were in..." consider deleting "of"

Thank you for indicating the mistake.

[9] Line 368: Here off-target effects are discussed as a possible explanation for the lack of distinct phenotypes resulting from DESAT RNAi knockdown. The above section on ELO (326-337) is more comprehensive in considering also substrate specificity and the possible shift in substrate concentrations along the entire pathway. It may be worth while to briefly mention these alternative explanations for DESAT too.

We approve the reviewer’s suggestion and added one sentence at the end of that paragraph: “The lack of distinct phenotypes and the difficulty to detect substrate specificity could be for the similar reasons as discussed above for elongases.”

[10] Line 446: It is interesting to know precisely how chemotype 1 and chemotype 2 differ chemically, and it may be helpful to the reader to be informed about this in the introduction, so that they can have it in mind that alkenes are particularly variable between chemotypes when reading the results section. Around line 99 it may be worth while to include a bit about the previously published finding that alkenes especially differ between chemotypes.

We thank the reviewer for this suggestion. We edited the sentence in the introduction and added two more sentences to provide the reader more details about the chemotype differences (lines 105–111):

“The two chemotypes (c1 and c2) qualitatively differ in 77 CHCs from each other, with most of the differing CHCs being alkenes¹⁴. Specifically, chemotype 1 is characterized by alkenes with double bonds at uneven positions, whereas chemotype 2 is characterized by alkenes with double bonds at even positions^{14,15}. Wasps expressing chemotype 2 furthermore feature a higher relative abundance of alkenes with chain lengths greater than 27 carbons than wasps expressing chemotype 1^{14,15} (Supplementary Figure 1).”

Note that we now also refer to Supplementary Figure 1 which illustrates these differences.

[11] Line 1042: If presenting a numbered list in the Table 1 legend, it may be best to provide a list with as many items as there are columns in the table, to avoid confusion. For example, in Table 2, there is a list of four items in the legend, corresponding to a table with four columns. But for Table 1 the list has four items and the table has five columns. Easily solved by adding "Wasp gene ID" as list item number 1 in the Table 1 legend.

We appreciate the comment and modified the legend accordingly: “Table 1. Cuticular hydrocarbon biosynthesis candidate genes in the mason wasp *Odynerus spinipes*. The table informs about 1) the mason wasp gene ID, 2) between which groups differential expression differences (DGE) were detected, 3) orthologous/homologous genes in the honey bee (*Apis mellifera*) genome, 4) the percentage of identical amino acids between the encoded sequences of the mason wasp and those of honey bee , 5) the genes’ predicted functions, and 6) the expression of the candidate genes in honey bee fat body cells.”

[12] Line 1078: The vertical scale bar, representing number of gene copies in collapsed clades, in Figure 2 may need to be longer. For example, the vertical height of the fan edge containing three Dipteran ELO C7 genes should, according to the scale bar, containing about 10 genes, if I am interpreting correctly. Line 1083: Please double check the scale bars here too.

We are grateful for the reviewer to have pointed our attention to the vertical scale. We missed to clearly indicate in the legend that the scale bar represented the number of gene copies per species. We modified it in the figure legends, and added one sentence in the Supplementary Material and Methods (lines 134 and 135) for better clarity:

“The width of each clade was computed as the number of gene copies per species within each clade.”

[13] Line 1099: Figure 5 legend, please include meaning of color code in PPLS-DA score plots

We added a description of the meanings of the colors in the legend.

[14] Line 1105, 1122: Figure 5 and 6 legends, minor typos: please change "correction" to "correlation" circle plot

We thank the reviewer for spotting this mistake: we corrected the legends.

[15] Line 1107: Figure 5 legend, please include meaning of color code in box plot

We added a description of the meanings of the colors in the legend.

[16] Supplementary PDF Line 90: searched the protein sets, not the genomes, is that right?

It is indeed correct, thank you for pointing that out. We modified the text accordingly.

[17] Supplementary PDF Line 423: minor typo: please change "correction" to "correlation" circle plot

We thank the reviewer for spotting this mistake: we corrected the legends.

Reviewer #2 (Remarks to the Author):

This manuscript by Moris et al. focuses on understanding CHC synthesis in two hymenopteran species: the mason wasp, *Odynerus spinipes*, and the honeybee, *Apis mellifera*. Based on their experiments, the authors conclude in their abstract “Our results set the base for experimental CHC profile manipulation in Hymenoptera and imply that the evolutionary origin of CHC biosynthesis predates the arthropods’ colonization of land.” I agree with the authors that the data presented in this paper contributed to experimental CHC profile manipulation in Hymenoptera. Even though as I agree with the authors’ claim that the evolutionary origin of CHC biosynthesis likely predates the arthropods’ colonization of land, this manuscript does not provide additional data beyond what is already known to support this.

We thank the reviewer 2 for his suggestions and analysis of our manuscript. We are glad to read that he agrees with our conclusion, but we are surprised by the reviewer’s statement that the manuscript would not provide data beyond what is already known to support the idea that the evolutionary origin of CHC biosynthesis likely predates the arthropods’ colonization of land. First, we have not come across a publication that has actually previously explicitly drawn our conclusion that the evolutionary origin of CHC biosynthesis likely predated the arthropods’ colonization of land. In our manuscript, we combined information on which genes are involved in CHC biosynthesis in *Apis mellifera* and in *Drosophila melanogaster*, referring to research conducted primarily by Wicker-Thomas. The combined information shows that in multiple gene families, multiple subclades, which must have diverged in an aquatic ancestor, occur. While our results are in accordance with the gene trees of fatty acid synthases and fatty acid elongases published by Finck et al., 2016, we base our conclusions on more experimental data (when subclade are indeed involved in CHC biosynthesis) and more gene family trees, which makes our hypothesis more reasonable.

We revised the discussion on the topic in order to make our claims clearer and to address the reviewer’s concerns (lines 477-484):

“Based on the four inferred gene trees and the ISH and knockdown results on *A. mellifera*, our study shows that all major gene families (*Figure 1*) and most subclades within them harboring genes known to be involved in CHC biosynthesis (i.e., FAS A, ELO C, ELO B, Desat A1, and potentially FAR C) are present in the extant major lineages of Euarthropoda. The phylogenetic relationships in the gene trees of fatty acid synthases and fatty acid elongases inferred by *Finck et al.*³² are consistent with this interpretation.”

We hope that the revised version of the corresponding paragraph in the discussion makes is more obvious what novel aspect we present in our manuscript.

This manuscript contains a lot of experiments and analyses. However, after reading this manuscript, I feel that this manuscript feels like at least two different manuscripts pieced together without a coherent message.

We agree that the manuscript contains a lot of experiments. We think presenting them all in a single manuscript present more advantages rather than splitting it into two manuscripts. Indeed, the dimorphism in *O. spinipes* females is mainly used as a tool for finding candidate genes that were tested in *A. mellifera*. The main aim of this study was to identify genes involved in CHC biosynthesis in Hymenoptera. We see the demand of better integrating the two parts (also suggested by the editor and by reviewer 3). We therefore rewrote the entire paragraph on *O. spinipes* in the

discussion in order to reduce speculations regarding the impact of the candidate genes in *O. spinipes* and better integrate this paragraph within the whole study (lines 520–559):

“RNAi experiments are difficult to conduct on *O. spinipes*, because *O. spinipes* cannot be cultured (at least so far), occurs at most known field sites in comparatively small numbers, and because rearing age-controlled specimens of this species is extremely time consuming (i.e., can only be done once per year). We therefore decided to assess *a posteriori* whether the possible functions of genes — inferred from our knockdown experiments on honey bees — that are differentially expressed in *O. spinipes* females with different chemotypes, can explain the specific phenotypic differences between the two chemotypes. This approach rests on the ortholog conjecture, which states that orthologous genes in different species tend to retain the same function⁴⁸. Intriguingly, two of the five genes that we found differentially expressed between *O. spinipes* females with different chemotypes (i.e., g14712, g7616) are likely sufficient to explain the majority of the phenotypic differences between the two chemotypes. Knockdown of GB51238, a co-ortholog of the *O. spinipes* fatty acid desaturase g14712, caused a decrease in the relative abundance of alkenes with double bonds at even chain positions in worker honey bees. Consistent with this result, we found g14712 to be higher expressed in *O. spinipes* females expressing chemotype 2, which is characterized by alkenes with double bonds at even chain positions (i.e., 8, 10, 12, and 14) than in *O. spinipes* females expressing chemotype 1, which is characterized by alkenes with double bonds at uneven chain positions (i.e., 7 and 9). Likewise, knockdown of GB51247, an ortholog of the *O. spinipes* fatty acid elongase g7616, caused a decrease of the relative abundance of CHCs with chain lengths larger than 27 carbon atoms in worker honey bees. Consistent with this result, we found g7616 to be higher expressed in *O. spinipes* females expressing chemotype 2, which is characterized by a high relative abundance of alkenes with chain lengths > 27 carbon atoms, than in *O. spinipes* females expressing chemotype 1, which is characterized by low relative abundance of alkenes with chain lengths > 27 carbon atoms (Supplementary Figure 1).”

We decided not to completely remove the discussion of what enzymes are involved in the manifestation of the CHC profile differences between the chemotypes, as it seems to be of interest for reviewer 1 and 3. We hope that the aims of our study have become clear after having revised the manuscript text and that the manuscripts now appears more coherent.

Major comments

1/ In this manuscript, the authors sequenced the genome of *Odynerus spinipes* and use transcriptomics to investigate which CHC synthesis genes may be involved in the two different CHC chemotypes in this species. They then perform in situ hybridization of CHC genes in *Apis mellifera* and performed RNAi knockdown of these genes in the honeybee. The authors then try to correlate experimental data in the honeybee to possible functions on these orthologous genes in the mason wasp. To me, this sounds like two different sets of experiments for two different manuscripts. While I have no problem with the authors doing this, based on the current set of experiments, some of the conclusions cannot be supported. CHC genes evolve rapidly across species, and this has been shown even within a single genus, *Drosophila*, as the elegant work of Claude Wicker-Thomas and colleagues have showed in the past few decades. The CHCs genes expressed in *Apis mellifera* may not be

expressed in similar cells as in *Odynerus spinipes*. This has been shown by Wicker-Thomas and colleagues that the expression genes like *desatF* and *eloF* evolved rapidly even within closely related species.

We highly appreciate the elegant work done by Wicker-Thomas and colleagues. They indeed showed that *eloF* and *desatF* are not expressed in *D. simulans* and that they were only found to be expressed in *D. melanogaster*. However, we think that experimental results from a single study system should not automatically be generalized. The null hypothesis of 1:1 orthologs is that these genes likely have retained the same function. Given the concerns raised by the reviewer, we revised our text and make your approach and assumption more explicit in the discussion (lines 520–534, already printed above):

“RNAi experiments are difficult to conduct on *O. spinipes*, because *O. spinipes* cannot be cultured (at least so far), occurs at most known field sites in comparatively small numbers, and because rearing age-controlled specimens of this species is extremely time consuming (i.e., can only be done once per year). We therefore decided to assess *a posteriori* whether the possible functions of genes — inferred from our knockdown experiments on honey bees — that are differentially expressed in *O. spinipes* females with different chemotypes, can explain the specific phenotypic differences between the two chemotypes. This approach rests on the ortholog conjecture, which states that orthologous genes in different species tend to retain the same function⁴⁸.”

In addition, there could be protein coding changes that changes substrate specificity of orthologues between species, and this is not taken into account.

We discuss substrate specificity of enzymes multiple times in the manuscript and acknowledge that a different suite of experiments is necessary to assess it. We nonetheless think that analyzing whether the expression of 1:1 orthologs in two species correlates with changes in the same compound class is legitimate.

Therefore, data on gene expression and RNAi knockdown in *Apis* cannot be conclusively transferred to *Odynerus*. Experiments that would be convincing would be to knockdown these genes in *Odynerus* or to express *Odynerus* genes in a cell culture system, similar to the Tupec et al 2019 *Elife* paper, which the authors failed to cite. This is perhaps the conceptual flaw in this manuscript.

We originally cited the nice manuscript from Tupec et al 2019 *Elife* and were well aware of it as the last author on our manuscript even reviewed it. It was cut out of our manuscript because of the restriction of the number of references. But perhaps an exception can be made and we could add it in the list. I provisionally added the citation in the text. We softened the discussion about *Odynerus*, as mentioned above. Unfortunately, RNAi experiments in *O. spinipes* are not possible at the moment, since this species cannot be cultured and cannot be collected in large quantities in the field. Generating cell cultures is a nice idea, and we will consider in for future projects that build on the results presented in our manuscript. We incorporated the reviewer’s ideas in the discussion (lines 320–324):

“In the light of these results, the remaining candidate genes not further experimentally assessed in the present investigation also appear highly promising targets for future reverse genetic and heterologous expression experiments, as done, for example, by Tupec et al.²¹ on fatty acyl-CoA reductases. “

2/The discovery that a fatty acid amide hydrolase affecting CHC profile composition is novel and promising but is buried by this manuscript's many different and diverse conclusions.

It is indeed very exciting result and we tried to emphasize on it as much as possible. However, it is not the only remarkable results. The other points discussed in the manuscript are also important. As mentioned before, it seems that fatty acid elongases and desaturases are well studied, but it is not the case. Our knowledge of what lineages within this gene families are involved in CHC biosynthesis rests primarily on research on *Drosophila*. Moreover, the localization of CHC biosynthesis is also important for future studies and the fact that trophocytes are also likely involved is important too. Finally, the evidence that the common ancestor of Euarthropoda likely possessed all genes required for CHC biosynthesis also had to be discussed. We significantly reduced the paragraph in the discussion on the genes' functions in *Odynerus spinipes* and hope that by doing so, it does not distract anymore from other important discoveries, such as the discovery of a FAAH gene impacting CHC profile composition. We now also present a gene tree of this gene family (requested by reviewer 3), which also gives the FAAH discovery more weight.

3/The authors hypothesized that the “capacity of CHC biosynthesis evolved before the divergence of the major lineages of Euarthropoda” is based on i) “genes known to be involved in CHC biosynthesis (FAS A, ELO C, ELO B, Desat A1, and potentially FAR C) are present in the extant major lineages of Euarthropoda” and that ii) “CHCs and oenocytes have also been reported from all major Euarthropoda lineages” . I agree with the authors, but the alternative hypothesis is that these genes are also involved in fatty acid synthesis for other endogenous functions and are present for that function. Oenocytes are also present in many species not just for CHC production but for other endogenous functions (Gutierrez et al. Nature 2007, Makki et al. Annual Review of Ent 2014).

We appreciate the reviewer's comment and reflection. Indeed, it is likely that genes, such as those encoding elongases, desaturases, fatty acid synthases, are involved in fatty acid synthesis and not (only) in CHC synthesis. However, we present in this manuscript the most parsimonious hypothesis based on multiple lines of evidence. Because CHCs and not only oenocytes (which can indeed have other functions than CHC production) have been found in all major Euarthropoda lineages and because the last ancestor already possessed genes utilized for CHC biosynthesis, the conclusion we draw is reasonable and the most parsimonious one. Yet, we understand the reviewer's point of view and therefore mention now explicitly the alternative hypothesis in our manuscript in lines 488–490:

“An alternative hypothesis, namely convergent exaptation of the same genes for CHC biosynthesis in different lineages of Euarthropoda, is far less parsimonious.”

Reviewer #3 (Remarks to the Author):

Cuticular hydrocarbons are important chemical signals in insects mediating various interactions among individuals. This manuscript investigates the genome and transcriptome of mason wasp to identify the candidate genes that mediate CHC dimorphism in this species. RNAi-based knockdown of the candidate gene orthologs was applied on honeybees, and their CHC profiles were compared with control individuals. Although the effects of RNAi on

target gene expression is controversial, most candidate gene modulates the CHC profiles as expected. Localization of candidate genes was also investigated, and some genes were expressed in trophocytes of the fat body. Generally, the study is interesting and contributes to our understanding of molecular bases of CHC diversity in insects, however as outlined in the comments below, the results of the manuscript are descriptive and the objective and conclusion of the study are obscure.

We thank reviewer 3 for his reading of our manuscript. We are glad to read that our study contributes to the understanding of the molecular bases of CHC diversity in insects. We revised the last paragraph of the introduction and the last paragraph of the discussion to make the objectives of the study clearer.

1. The communications via CHCs have important roles in social Hymenoptera, and identifying genes involved in CHC biosynthesis of Hymenoptera is, indeed, of major interest (Line 74-80).

In this study, the authors identified several elongase and desaturase as genes involved in CHC profiles. The product of these gene families is well-known as biosynthetic enzymes for CHCs and several pheromones in non-Hymenopteran insects. In both families, multiple genes from multiple clades were involved in CHC biosynthesis. Are there any characteristics in each clade, such as the Hymenoptera-specific diversification pattern? The authors discussed the evolution of CHC biosynthesis genes Euarthropoda level, but the evolution of gene repertoire at more local levels (i.e. family levels) will be needed for this manuscript.

We thank the reviewer for this comment. We first want to emphasize that if elongase and desaturase genes are often cited as “well-known biosynthetic enzymes for CHCs”, not many genes have been clearly identified as being involved in CHC biosynthesis. Indeed, only three desaturases have been shown to be involved in CHC biosynthesis and exclusively in *Drosophila melanogaster* (reviewed by Holze et al. 2020). These genes are very likely not representative for all insects or, more specifically, for Hymenoptera. So far, no desaturase gene has been found to add a double bond at an even position either. Therefore, there is still a lot to discover about this gene family, and our manuscript significantly improves knowledge about it. Likewise, there is no information on what specific fatty acid elongases impact CHC abundance in Hymenoptera. Because of this remark, though, we realized that we should better emphasize this fact in the introduction, and we modified two sentences accordingly (lines 87-95):

“However, the details of CHC biosynthesis and which genes encode the participating enzymes are not yet fully elucidated, particularly in the species-rich insect order Hymenoptera, in which only a single gene (CYP4G) has so far been shown to be involved in CHC biosynthesis^{10,11}. Furthermore, insights obtained from studying species of one insect lineage (e.g., Diptera, in which three genes coding for fatty acid desaturases have been shown to be involved in alkene synthesis based on research on *Drosophila*¹⁰) are not necessarily representative for species of a distantly related lineage (e.g., Hymenoptera).”

Concerning the evolution of these gene families in Hymenoptera, it was simply not the aim of our study. To answer this question properly, one would have to include the proteomes of many more species to obtain a large number of independent contrasts from valid statistical analysis. However, this remark by reviewer 3 and another one by reviewer 1 made us realize that the vertical scales in the gene trees are not well annotated. The presented scale represents the number of gene copies per species. This allows us to assess whether there is a major radiation event in one clade, such as

Hymenoptera. We did not find Hymenoptera to stand out in this regard from other insect orders. Yet, since the gene tree represents the number of gene per Euarthropod clade and not per species, we thought it that it would indeed be interesting to give more details in the discussion about the number of copies of each gene family in each species and to determine if specific-species radiation events occurred. Therefore, we made a new *Supplementary Figure 9* that shows the number of gene copies per species in each gene family (including FAAH) and added a new paragraph to discuss it (lines: 497–508):

“Although CHCs serve social insects as important semiochemicals (e.g., cast and nest recognition), social species do not possess more complex CHC profiles than solitary ones⁴³. CHCs indeed also serve solitary species fundamental functions (e.g., cues and signals for species and sex recognition²). We found the absence of systematic differences between social and solitary species in their CHC profiles to be reflected at the genetic level: none of the taxonomic lineages represented in the gene trees stands out by a particularly rich or poor set of genes, irrespective of whether or not the lineage contains solitary or social species (*Figures 2 and 3; Supplementary Figures 2 and 3*). What we found are species-specific gene duplications in *Amyelois transitella*, *Manduca sexta*, *Nasonia vitripennis*, and *Tribolium castaneum* (*Supplementary Figure 9*), possibly linked to species-specific behaviors other than sociality.”

Finally, in order to make the gene trees more accessible for additional analyses by readers, we now provide them as treefiles in the supplementary information on zenodo. With these files, other researcher can look for specific species or genes within the gene trees.

In addition to these genes, a fatty-acid amide hydrolase is a novel CHC biosynthesis gene identified in this study. Does this gene have any important function for CHC signaling in Hymenoptera?

We are the first to report its involvement in CHC biosynthesis in Hymenoptera, therefore no functional information is available at the present moment.

How diversified is this gene family in insects? Despite its importance and novelty, the analysis of this gene is insufficient.

We took the note as an occasion to search amongst all studied genomes of 37 species of Euarthropods genes with the amidase domain which is characteristic of the gene candidate g2290 of *O. spinipes* and the RNAi target gene of *A. mellifera*. The gene tree inferred from the identified gene family members is provided as a new *Supplementary Figure 8*. Based on the hmmer search performed for this gene family, we found 2 to 9 copies in insect genomes. As we studied only a single member of the gene family via gene knockdown, it is at this point impossible to provide more information on how important this gene family is in shaping CHC profiles and what enzymatic reactions the encoded enzymes catalyze. However, we deem reporting the exciting discovery important and are confident that it will trigger future experiments. We added a sentence in the manuscript (lines 444–447) in which we provide this information:

“It is likely that another honey bee gene phylogenetically closer to the candidate gene g2290 (GB48850) is also involved in CHC biosynthesis (*Supplementary Figure 8*). Future experiments (e.g., heterologous expression) should shed light on the role of this small gene family in CHC biosynthesis.”

We also added the number of gene copies in each studied species in the new Supplementary Figure 9.

Overall, although the importance of Hymenopteran CHCs is emphasized in the Introduction, much of the discussion is limited to functional descriptions of individual genes in insects.

We thank the reviewer for this comment. The discussion is indeed about the description of individual genes that we identified in *Apis mellifera*, a representative species of Hymenoptera. Except for a single study, which identified a cytochrome p450 oxidative decarboxylase (Calla et al., 2018), this is the currently the only study that experimentally identified genes that modulate CHC profile composition in Hymenoptera. Our results will allow other researchers to better target genes involved in CHC biosynthesis in their research on Hymenoptera, to manipulate CHC profiles in species of this mega-diverse insect order, to study the effects for CHC manipulation on insect behavior. We added a sentence to stress these and other deliverables of our study at the end of the discussion (lines 576–580):

“We presented an array of genes that impact CHC profile composition in Hymenoptera. These enable future experimental manipulation of CHC profiles in this mega-diverse insect order and functional investigations on the activity and substrate specificity of the encoded proteins.”

2. The molecular bases of two distinct CHC phenotypes in *O. spinipes* is another (sub) topic of this study. It is necessary to explain more about previous studies of this species.

We agree and added a few lines in the introduction to give more information about previous studies concerning the two chemotypes of *O. spinipes* females (also mentioned by reviewer 1).

Although the mitochondrial haplotypes have not differed between the two phenotypes (Wurdack et al., 2015 Proc. R. Soc. B 282 20151777), it is still possible that the two phenotypes are genetically determined not in the transcript levels. Even if *O. spinipes* females with different chemotypes do not appear to differ in any other trait from each other (Line 102-103), the expression of the CHC biosynthesis gene is not the only possible explanation. Are genome sequencing, assembly, and annotation of mixed male and female DNA valid, even when the phenotype is genetically determined? Please explain and discuss the possible mechanisms of CHC phenotypes in this species.

We thank the reviewer for this interesting comment and his/her interest in this extraordinary dimorphism. When the dimorphism was discovered, we first had to verify that we indeed do not deal with two reproductively isolated species (see Wurdack et al., 2014; Moris et al., 2021). We were able to refute this possibility. The next step was to see whether individuals of the species are able to phenotypically switch between the two chemotype during the wasp’s lifespan. We were able to refute this possibility with data presented in the current manuscript. However, it is not the topic of our study to assess how the dimorphism is genetically controlled (e.g., by a super gene construct). This is an entirely different line of research that we pursue and will hopefully be able to present in a future manuscript dedicated to this research question. Also note that it is indeed possible that the two chemotypes differ also in genes not associated with biosynthesis of CHCs. As we compared the transcriptomes only to identify candidate genes that we subsequently experimentally verified, the approach bears no risk of falsely reporting genes that do not modulate CHC profile composition. We addressed the question whether samples used for genome

sequencing had the same chemotype by adding a corresponding sentence in the supplementary methods (line 49).

I suggest that this manuscript is better presented as two separate papers so that the two topics (one for wasps and the other for honeybees) can be discussed clearly and in detail.

We understand the wish of the reviewer to discuss separately the establishment of the dimorphism in *O. spinipes*. And as outlined above, this question will be dealt with in a future manuscript. The current manuscript focuses on the identification of genes involved in CHC profile modulation in Hymenoptera. *O. spinipes* was solely used to identify candidate genes which were then experimentally further assessed in *A. mellifera*. The dimorphism of *O. spinipes* helped us to choose which genes to target first in large gene families (as we observed for instance for fatty acid elongases), in which it would otherwise be hard to choose which ones to target in experiments first. Moreover, this strategy made the discovery of an enzyme (FAAH) previously not associated with CHC biosynthesis possible in the first place.

If we would remove from this manuscript how we identified our candidate genes, we think the manuscript loses in coherence. Moreover, the fact that most candidate genes identified by in *O. spinipes* modulate CHC profile composition also in the comparatively distantly related honey bee in our opinion increase the strength of our manuscript, as it shows that the genes have likely retained their ability to modulate CHC profile composition over evolutionary time. Keeping these two parts together also allows us to make some assumptions about how the genes targeted in *A. mellifera* act in shaping (but not controlling) the two CHC profiles expressed in *O. spinipes*.

Because two reviewers made this remark, we modified the last paragraph of the introduction to make the aims of our study clearer in the manuscript.

Minor comments:

Line 239-241: Please show the results of GB51236 as the figure, at least in supplements.

We added the results as Supplementary Figure 4 and changed the numbers of the other Supplementary Figures accordingly.

Figure 5-6: In score plots, both axes indicate the principal components calculated by PCA? If not (i.e. it is a score of DA), a simple "component 1" or "score 1" will be better to avoid confusion.

The information is provided at each each row and column. We refrained from redundantly provide the information at each subplot to increase and hence the readability of the composite figures.

Figure 7: I couldn't understand why the asterisk indicates the significant abundant differences. Y-axis is indicated as chain length. Then, why the chain lengths are so different among the control of each gene treatment? Is that mean the injection of GFP dsRNA also modulate the chain length of CHCs?

The chain length values (y axis) in RNAi treatments and in control groups are calculated from the CHCs that differed in the discriminant analysis between the two

groups. As the individual experiments targeted different genes that affected the abundance of different CHCs, there is no reason why the values of the control groups should be similar across

experiments. We added more details in the Methods and Material section: lines 788–795. We also added the CHC plots on zenodo ([zenodo: 10.5281/zenodo.5552394](https://zenodo.org/doi/10.5281/zenodo.5552394)) where we can better see which CHCs are positively correlated with the principal component 1 and the threshold.

REVIEWERS' COMMENTS:

Reviewer #1 (Remarks to the Author):

The authors have in my opinion fully addressed my minor concerns and recommendations.

2 minor typos in the methods section:

line 575 - should be "Shimadzu"

line 592 - should be "Hong Kong, PRC"

The article presents new results identifying cuticular hydrocarbon biosynthesis genes by leveraging the intrasexual chemotype dimorphism of a wasp and the molecular genetic tractability of the honey bee.

Reviewer #2 (Remarks to the Author):

Reviewer 2 here. I appreciate the authors' replies to my comments on their manuscript. I acknowledged that RNAi experiments are very difficult to do in many species and it wasn't my original intent to ask the authors to do those experiments in *O. spinipes*. However, the authors have to cite the Tupac et al paper in the discussion saying that expressing their candidate genes in a cell culture system can provide more information about the functions of these genes.

I agree with the authors that "the capacity of CHC biosynthesis evolved before the divergence of the major lineages of Euarthropoda", with "capacity" being the operative word here. I still disagree on the authors' viewpoints that we can attribute the functions these genes to CHC synthesis simply based on conservation/orthology. "The null hypothesis of 1:1 orthologs is that these genes likely have retained the same function" probably holds true for many genes involved in essential functions but CHC synthesis genes evolved really rapidly (desatF across *Drosophila* species, Shirangi et al. for example) so I would not put my money on it. However, I think the authors are free to make their conclusion if they wish (this is their paper and not mine), and perhaps time and additional research in the future will give us more information about this topic.

Congratulations to the authors for completing this nice manuscript which contributes a lot to the CHC literature.

Reviewer #3 (Remarks to the Author):

Thank you for your responses to my comments. The MS is now appropriately revised. In particular, I believe additional analyses and paragraphs for the number of gene copies (Supplementary Figs 8 and 9) make the MS better for a range of insect researchers.

REVIEWERS' COMMENTS:

Reviewer #1 (Remarks to the Author):

The authors have in my opinion fully addressed my minor concerns and recommendations.

2 minor typos in the methods section:

line 575 - should be "Shimadzu"

line 592 - should be "Hong Kong, PRC"

The article presents new results identifying cuticular hydrocarbon biosynthesis genes by leveraging the intrasexual chemotype dimorphism of a wasp and the molecular genetic tractability of the honey bee.

Thank you for spotting these two remaining misspellings and your comments.

Reviewer #2 (Remarks to the Author):

Reviewer 2 here. I appreciate the authors' replies to my comments on their manuscript. I acknowledged that RNAi experiments are very difficult to do in many species and it wasn't my original intent to ask the authors to do those experiments in *O. spinipes*. However, the authors have to cite the Tupac et al paper in the discussion saying that expressing their candidate genes in a cell culture system can provide more information about the functions of these genes.

Following the first review, we already cited Tupec et al. (line 318). However, following this suggestion of the reviewer, we added another citation of the study another time in the paragraph dedicated on *O. spinipes* in the discussion (lines 509-512).

“Heterologous expression experiments²¹ could provide the means to test our above hypotheses and to shed light on the specific enzymatic reactions catalyzed by the remaining identified candidate genes in *O. spinipes*.”

I agree with the authors that “the capacity of CHC biosynthesis evolved before the divergence of the major lineages of Euarthropoda”, with “capacity” being the operative word here. I still disagree on the authors' viewpoints that we can attribute the functions these genes to CHC synthesis simply based on conservation/orthology. “The null hypothesis of 1:1 orthologs is that these genes likely have retained the same function” probably holds true for many genes involved in essential functions but CHC synthesis genes evolved really rapidly (desatF across *Drosophila* species, Shirangi et al. for example) so I would not put my money on it. However, I think the authors are free to make their conclusion if they wish (this is their paper and not mine), and perhaps time and additional research in the future will give us more information about this topic.

Thank you for your comment. We think with the additional sentence (mentioned above, lines 509-512) indicates that these interpretations could be tested via other methods such as heterologous expression experiments.

Congratulations to the authors for completing this nice manuscript which contributes a lot to the CHC literature.

Thank you for this comment, we are happy to contribute to the CHC literature.

Reviewer #3 (Remarks to the Author):

Thank you for your responses to my comments. The MS is now appropriately revised. In particular, I believe additional analyses and paragraphs for the number of gene copies (Supplementary Figs 8 and 9) make the MS better for a range of insect researchers.

Thank you for this comment.